# CD5L as a promising biological therapeutic for treating sepsis

Liliana Oliveira [1,2], M. Carolina Silva [1,2,3], Ana P. Gomes[1,2], Rita F. Santos[1,2,4], Marcos S. Cardoso [1,2,4], Ana Nóvoa[5], Hervé Luche [6], Bruno Cavadas[1], Irina Amorim [1,7,8], Fátima Gärtner[1,7,8], Bernard Malissen [6], Moisés Mallo [5] & Alexandre M. Carmo [1,2] ✉

Sepsis results from systemic, dysregulated inflammatory responses to infection, culminating in multiple organ failure. Here, we demonstrate the utility of CD5L for treating experimental sepsis caused by cecal ligation and puncture (CLP). We show that CD5L's important features include its ability to enhance neutrophil recruitment and activation by increasing circulating levels of CXCL1, and to promote neutrophil phagocytosis. CD5L-deficient mice exhibit impaired neutrophil recruitment and compromised bacterial control, rendering them susceptible to attenuated CLP. CD5L$^{-/-}$ peritoneal cells from mice subjected to medium-grade CLP exhibit a heightened pro-inflammatory transcriptional profile, reflecting a loss of control of the immune response to the infection. Intravenous administration of recombinant CD5L (rCD5L) in immunocompetent C57BL/6 wild-type (WT) mice significantly ameliorates measures of disease in the setting of high-grade CLP-induced sepsis. Furthermore, rCD5L lowers endotoxin and damage-associated molecular pattern (DAMP) levels, and protects WT mice from LPS-induced endotoxic shock. These findings warrant the investigation of rCD5L as a possible treatment for sepsis in humans.

CD5 antigen-like (CD5L), also known as apoptosis inhibitor expressed by macrophages (AIM), is a protein found circulating in the bloodstream. CD5L serum levels vary significantly in several pathologies, which led to consider this molecule as a diagnostic or prognostic marker in different clinical scenarios, from infection, cancer, and autoimmune diseases[1–10]. However, while the precise average total CD5L concentration in the serum of healthy human donors was determined to be 60 µg/ml through proteomics methods[11], the considerable variability in individual studies, where values ranged from 0.1 to 60 µg/ml, and the utilization of distinct enzyme-linked immunosorbent assay (ELISA) kits or alternative methodologies render cross-study comparisons nearly impractical and pose challenges in assessing consistency[3,10,12]. Furthermore, serum values may not provide much clarification on the biological role of CD5L, as most quantitation assays do not distinguish between biologically inactive CD5L, which is complexed with serum immunoglobulin M (IgM), and the free, immunologically active form of CD5L[12–15].

In health, most circulatory CD5L is covalently bound to pentameric IgM to form a stable complex in which both components reciprocally regulate the activity of one another[11,16]. Association with CD5L diminishes IgM binding to Fcαμ and polymeric immunoglobulin receptors, inhibiting, for example, IgM immune complexes from

[1]i3S - Instituto de Investigação e Inovação em Saúde, Universidade do Porto, Porto, Portugal. [2]IBMC - Instituto de Biologia Molecular e Celular, Porto, Portugal. [3]Universidade de Aveiro, Aveiro, Portugal. [4]ESS, Politécnico do Porto, Porto, Portugal. [5]Instituto Gulbenkian de Ciência, Oeiras, Portugal. [6]Centre d'Immunophénomique (CIPHE), Aix Marseille Université, INSERM, CNRS, 13288 Marseille, France. [7]ICBAS - Instituto de Ciências Biomédicas Abel Salazar, Universidade do Porto, Porto, Portugal. [8]IPATIMUP - Instituto de Patologia e Imunologia Molecular da Universidade do Porto, Porto, Portugal. ✉e-mail: acarmo@ibmc.up.pt

receptor-mediated internalization, while IgM-complexed CD5L is biologically inert. Under different disease conditions, however, a fraction of CD5L dissociates from IgM and becomes an autonomous biologically active component of the immune system[14,15].

Also, it is at present undetermined whether CD5L is a protective immune factor that combats disease or, instead, that it may contribute to aggravating deleterious responses. In fact, the anti-apoptotic role of CD5L, for which the molecule was initially recognized, may actually contribute to divergent immune responses[17]. Contingent on the type of cells rescued from programmed death, CD5L can either contribute to fighting infection and helping to resolve inflammation, or on the contrary be detrimental to disease progression. An example of the former was provided by a mouse model of *Listeria monocytogenes* infection, where CD5L promoted the survival of infected macrophages and improved antimicrobial functions[18]. Conversely, in a mouse model of atherosclerosis, CD5L-mediated survival of macrophage-derived foam cells resulted in increased macrophage-driven inflammation and more advanced atherosclerotic lesions[19].

CD5L also acts as a pattern recognition receptor (PRR), which is a hallmark of the scavenger receptor cysteine-rich (SRCR) family to which CD5L belongs. Like the circulating SRCR molecules SSC4D[20], SSC5D[21], and DMBT1[22], CD5L can recognize and bind pathogen-associated molecular patterns (PAMPs) of microbial cells, like bacterial lipoteichoic acid (LTA) and lipopolysaccharide (LPS), or fungal mannan and β-D-glucan[23,24]. This opsonization-like mechanism may promote or facilitate phagocytosis of whole microorganisms[25,26].

Other impacting characteristics of CD5L include its anti-inflammatory and healing roles. Indeed, the interaction of CD5L with particular microbial cell wall components during pathogen-neutralizing reactions provides signals to immune cells to alleviate inflammation. This can be translated into reduced macrophage secretion of tumor necrosis factor (TNF)-α, interleukin (IL)−1β, and IL-6[23,24,27], restraining the pro-inflammatory signature of non-pathogenic Th17 cells[28], or, as observed in asthmatic mice, by inhibiting NLRP3 inflammasome activation and consequently reducing airway inflammation and Th2 immune responses[29]. Also, by binding to damage-associated molecular patterns (DAMPs) from host necrotic or dead cells, CD5L helps to clear cellular debris, contributing to decreasing the sterile inflammation associated with these forms and serving as a facilitator of tissue repair in different in vivo models[30–34]. However, it should be noted that there are also examples where CD5L may worsen inflammation, like in a model of LPS-induced lung injury, where CD5L promoted the occurrence of an "inflammatory/pathologic" lipid profile and delayed the resolution of inflammation[35].

Sepsis is fundamentally an inflammatory disease, a clinical syndrome related to an inadequate response to infection, and defined as multiple organ failure associated with an infection[36]. It accounts for nearly 20% of all deaths worldwide[37], although this is not perceived by the general public perhaps because sepsis often arises as a complication of underlying comorbid conditions, including cancer and chronic organ dysfunctions. Sepsis is also a leading cause of in-hospital and in-intensive care unit (ICU) deaths[37]. Once diagnosed, the mortality in less severe cases is estimated at ~10%, but it can reach a staggering 40-50% if it progresses to septic shock. A recent study reported that patients with sepsis at admission in ICUs had markedly increased serum CD5L levels, compared with ICU non-sepsis patients and with healthy subjects[38]. Within sepsis patients, serum CD5L levels were significantly higher in patients with septic shock compared to patients without shock, and in patients with bacteremia compared with those without bacteremia. Furthermore, serum CD5L levels correlated significantly with leukocyte counts, alanine aminotransferase (ALT) and procalcitonin (PCT) levels, and sequential organ failure assessment (SOFA) scores.

Given the uncertainty of the role that CD5L has in infection and sepsis, we established mouse models to analyze both susceptibility

to disease in the absence of this circulatory factor, and also the potential benefits of administering recombinant CD5L (rCD5L) to fight the experimentally inflicted disease. For this strategy, we initially developed mice with an inactivated *Cd5l* gene and subjected them to attenuated infectious or inflammatory insults; then, to evaluate to which extent the therapeutic administration of rCD5L could be beneficial or harmful, we provoked the full-scale pathology in wild-type (WT) animals and subsequently treated them with rCD5L. The disease models used were the cecal ligation and puncture (CLP) polymicrobial model of sepsis and the LPS model of endotoxic shock. Although LPS-induced shock is not a de facto sepsis model because there are no foci of infection, and CLP, while a genuine and prototypical sepsis model, is illustrative only of sepsis with abdominal origin, the uncontrolled inflammation caused in response to the endotoxin or the polymicrobial infection can give some insight into the process of sepsis[39,40]. Importantly, these two models cover more than 80% of studies addressing experimental sepsis in recent decades, making them a powerful tool for comparing the specific characteristics and efficacy of putative therapeutics across a wide range of preclinical treatments.

Here, we report that therapeutic administration of rCD5L rescues an unprecedented proportion of C57BL/6 mice from death caused by CLP or LPS, making this recombinant protein one of the most effective therapeutic agents in experimental animal models of sepsis and septic shock.

## Results

### CD5L-deficient mice have impaired immune responses and decreased survival in an otherwise non-lethal CLP model

CD5L-deficient mice (hereafter referred to as CD5L⁻ mice) were generated by CRISPR/Cas9 engineering by targeting the SRCR domain 1-encoding exon 3 of the *Cd5l* gene in C57BL/6 mice, with the insertion of three in-frame stop codons and a frameshift (Fig. S1a). CD5L was not detectable in the blood of CD5L⁻ animals, in contrast to the high levels of circulating CD5L in WT mice, as measured by ELISA (Fig. S1b). Tissue macrophages are the main source of CD5L[17]. Accordingly, peritoneal F4/80⁺ macrophages of WT mice expressed high levels of the protein, as detected by immunofluorescence. In contrast, equivalent cells from CD5L⁻ mice completely lacked CD5L production (Fig. S1c). CD5L⁻ mice were viable and healthy, and the analysis of leukocyte numbers (Fig. S2) and frequency of cell subsets (Fig. S3) from the spleen, thymus, peripheral blood, and peritoneal cavity at 12 weeks of age showed no significant differences between WT and CD5L⁻ mice.

CD5L deficiency resulted in high susceptibility of the mice to a medium-grade CLP surgical procedure, as shown by more than 60% of CD5L⁻ mice succumbing to disease, compared with 100% survival of WT mice subjected to a similar process (Fig. 1a). Furthermore, CD5L⁻ mice showed greater weight loss compared with WT mice, indicating a worse general condition (Fig. 1b). CD5L⁻ mice had elevated blood bacteremia 72 h after CLP, in contrast to the very effective control of systemic infection by WT mice [more than 10,000-fold fewer colony-forming unit (CFU) counts in the blood than CD5L⁻ mice] and, correspondingly, CD5L⁻ mice also had significantly higher bacterial counts in lungs, liver and kidneys than WT controls (Fig. 1c).

Leukocyte recruitment to the site of infection was increased in both WT and CD5L⁻ CLP-challenged mice, compared with sham-operated animals, but was delayed in the CD5L⁻ mutants (Fig. 1d). CD5L⁻ mice showed a significantly reduced number of recruited neutrophils than WT mice, but there were no differences in the numbers of monocytes/macrophages recruited into the peritoneal cavity. CD5L has been reported to be involved in the acquisition of an M2 phenotype by macrophages[27], and thus we assessed a potential imbalance between pro-inflammatory and anti-inflammatory phenotypes within the peritoneal macrophage population, using the classical markers

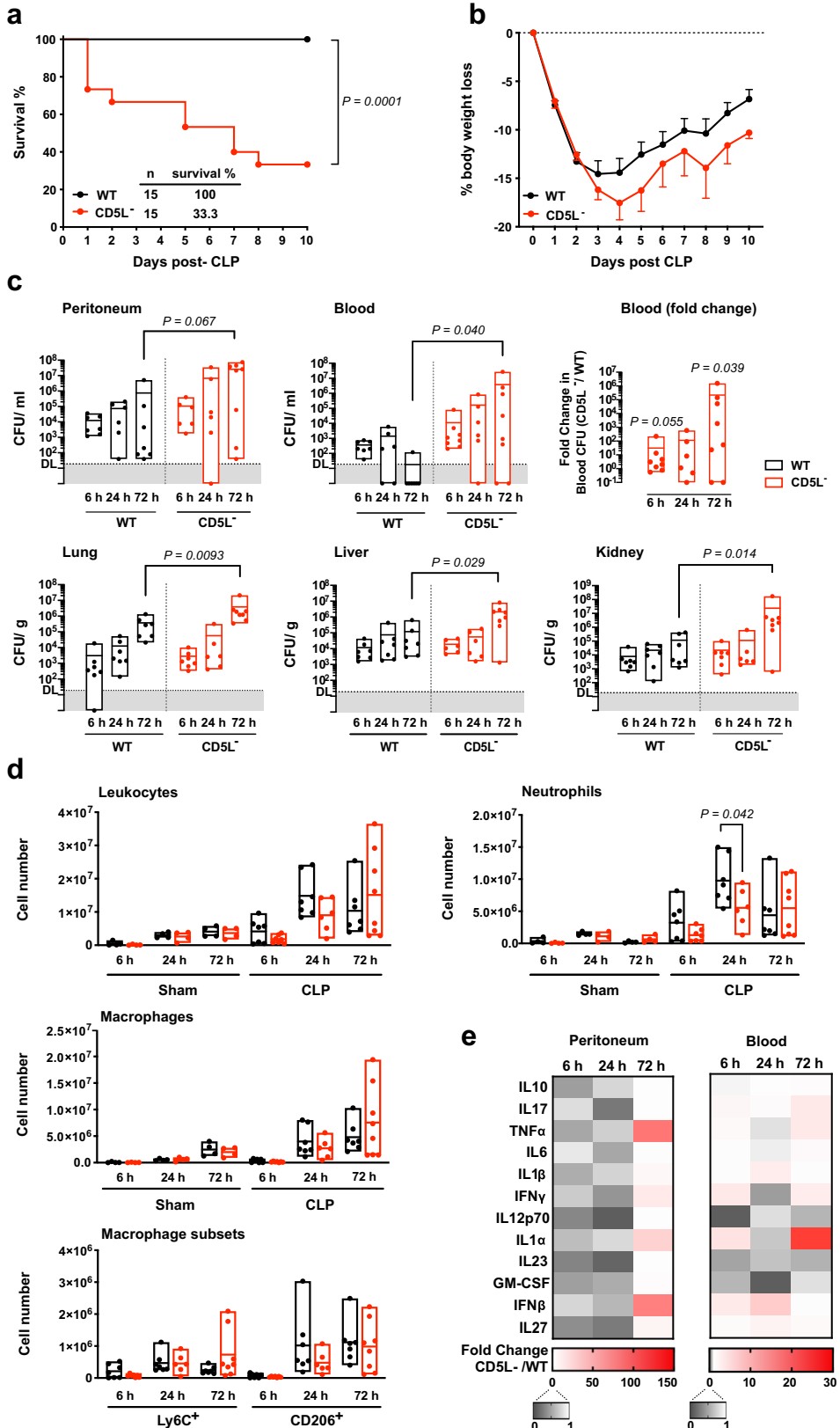

Ly6C and CD206, respectively. However, we found no significant differences in these M1- and M2-like profiles in the infiltrating populations, between WT and CD5L⁻ mice.

A multiplex analysis of key inflammation-related cytokines in peritoneal fluids and blood showed differences, between WT and CD5L⁻ mice, in only a few cytokines (Fig. S4), but the inflammatory response was faster in WT mice (6 – 24 h) and slightly delayed in CD5L⁻ mice, most evidently observed in the peritoneal cavity 72 h after midgrade CLP (Fig. 1e).

Histopathological analysis of the lungs, liver, and kidneys showed no relevant differences between WT and CD5L⁻ samples. With the exception of the lung, where microscopic changes were observed

**Fig. 1 | CD5L⁻ mice have a compromised immune response to mid-grade CLP.** WT and CD5L⁻ mice were subjected to cecal ligation and puncture (CLP) surgery to induce moderate disease severity. **a** Kaplan–Meier survival curves were generated to compare survival between the two groups and significance was determined by log-rank (Mantel-Cox) test. Graphical representation of pooled individuals ($n = 15$ in each group) from 5 independent experiments. **b** Quantification of body weight loss in the experimental setting described in **a**. Data are presented as mean values ± SEM. **c**–**e** WT and CD5L⁻ mice were subjected to mid-grade CLP; control mice (sham) underwent the same surgical procedure but without ligation and puncture of the cecum. Mice were sacrificed 6, 24, or 72 h after surgery. Pooled data from at least 2 independent experiments. **c** CFU counts of the indicated tissues from WT and CD5L⁻ mice after mid-grade CLP. Statistical differences between groups were analyzed by two-tailed Mann-Whitney test. DL, detection limit = 20 CFU. Mice per group at 6, 24, and 72 h: Peritoneum WT: 7, 6, and 7; peritoneum CD5L⁻: 6, 6, and 8. Blood WT: 7, 6, and 7; blood CD5L⁻: 8, 6, and 8. Lung, liver, and kidney WT: 7, 7, and 7; lung, liver and kidney CD5L⁻: 7, 6, and 8. **d** Absolute number of total CD45⁺ leukocytes, CD45⁺CD11b⁺Ly6G⁺ neutrophils, CD45⁺CD11b⁺CD11cˉF4/80⁺ macrophages and Ly6C⁺ and CD206⁺ subsets within macrophage population, assessed by flow cytometry of cell populations in the peritoneal cavity. Statistical comparisons were drawn after performing two-tailed unpaired t-tests with Welch's correction. Mice per group at 6, 24, and 72 h: Sham, WT, and CD5L⁻: 4, 4, and 4. CLP WT: 7, 7, and 7; CLP CD5L⁻: 7, 6, and 8. Floating bars show the minimum, average (line), and maximum values within each group (**c-d**). **e** Fold change in the indicated cytokines between CD5L⁻ and WT mice, quantified by bead-based multiplex immunoassay in samples from the peritoneal cavity (left panel) and blood serum (right panel).

somewhat earlier in CD5L⁻ mice, pathology scores were comparable between WT and CD5L⁻ organs (Fig. S5a, b). Nevertheless, serum aspartate aminotransferase (AST) values, measured 72 h after mid-grade CLP, were higher in CD5L⁻ mice (Fig. S5c). Renal dysfunction, assessed by serum creatinine levels and pathological score, was similar between the two mouse strains (S5a–c). There were also no significant differences in the levels of the main cytokines, measured by real-time qPCR (RT-qPCR), in the different organs 6 – 72 h after mid-grade CLP, except for an early (6 h) increase in tumor necrosis factor (TNF)-α values in the kidneys of CD5L⁻ mice, compared to WT, but these were leveled at later time points (Fig. S5d).

Our results unequivocally show that the presence of endogenous CD5L contributes to containing bacterial dissemination and confers total protection against acute infection, resulting in the absolute survival of the animals. There were some signs, although not exuberant, of impaired inflammatory responses in CD5L⁻ mice, but it was not entirely clear which of CD5L's many properties, immunological, antimicrobial, anti-inflammatory, or other, contributed to the protection. Thus, we analyzed transcriptomic signatures of the cell population that conferred immune protection locally at the site of infection, by performing bulk RNA-sequencing (RNA-seq) of cells recovered from the peritoneal cavity at 0 and 6 h after mid-grade CLP in WT and CD5L⁻ mice.

Principal component analysis (PCA) of all expressed genes showed a clear separation between healthy mice and those subjected to infection but with little distinction between WT and CD5L⁻ mice (Fig. 2a). Differential gene expression analysis revealed, upon mid-grade CLP, 507 genes upregulated and 133 genes downregulated in CD5L⁻, compared to WT mice (Fig. 2b). To identify gene sets with similar biological activity with significant changes in transcriptomic levels in CD5L⁻ peritoneal cells, we employed a gene set enrichment analysis (GSEA) using the MSigDB mouse hallmark gene sets (Fig. 2c). Genes having the highest variation within the top 7 up-regulated pathways are highlighted in Fig. 2d.

CD5L⁻ cells showed an increase in transcripts encoding proteins involved in IFN-α and IFN-γ responses and in TNF-α signaling which, together with an increase in the inflammatory response, suggests an immune response geared toward combating the polymicrobial infection[41–43]. Other transcripts enriched in CD5L⁻ cells during mid-grade CLP are involved in IL-6/JAK/STAT3 signaling, which is associated with inflammation and immune responses[44,45]; hypoxia, indicative of cellular stress[46,47]; and apoptosis, which indicates that the control of programmed cell death is necessary for the biological reactions that occur in the peritoneal cavity during CLP[43,48]. The combination of enriched interferon pathways, increased inflammatory response, IL-6/JAK/STAT3 signaling, hypoxia, and apoptosis seen in the transcriptomic analysis may signify a complex interplay between immune activation, increased inflammation, cellular stress, and potential elimination of affected cells, consistent with the complex and dysregulated immune response observed in sepsis.

Other transcripts upregulated in CD5L⁻ peritoneal cells, including *Osm, Il18bp, Ripk1, Nub1, Tlr2, Stat1, Irf1, Nfkb1, Pik3r5*, or their coding proteins (Fig. 2d), were already associated with sepsis severity[49–57]. On the other side of the spectrum, the downregulation of oxidative phosphorylation pathways may indicate a more severe inflammatory context (Fig. 2c). Although still controversial, inflammation-induced mitochondrial disfunction is being increasingly recognized in sepsis, and decreased ATP levels were described in both human subjects and animal sepsis models[58–61]. Overall, these data globally suggest that CD5L has a fundamental role in controlling inflammation during acute infection caused by mid-grade CLP.

## CD5L administration significantly decreases mortality upon lethal CLP in WT mice

Given that CD5L⁻ mice had impaired control of bacterial spreading and decreased survival than WT mice in the mid-grade CLP model, we explored whether administration of the recombinant mouse CD5L (rCD5L) protein could synergize with endogenous CD5L levels to combat the overt infection and inflammation likely to develop following a high-grade CLP procedure. The safety and tolerability of rCD5L were tested in naïve WT mice, injected intravenously with 2.5 or 5.0 mg/kg of rCD5L, or with PBS alone. No adverse effects were observed as mice remained healthy, AST and creatinine levels were maintained in line with vehicle-alone conditions (Fig. S6a), and microscopic analysis of the lungs, liver, and kidneys showed no relevant histopathological alterations (Fig. S6b, c).

We established two therapeutic protocols, administering rCD5L either intraperitoneally (IP), therefore directly to the infection site, or via intravenous (IV) injection, a more relevant procedure with a view to future translation into human medicine. The protocols were also designed to replicate clinical care as recommended for preclinical studies, taking into account the expected delays between diagnosis and initiation of treatment[62]. Consequently, therapy was initiated in WT mice with a 3-h delay after CLP for the first rCD5L dose [2.5 mg/kg (∼ 50 μg)], followed by a subsequent dose administered at 6 h. The control groups received PBS alone at the same time points. Survival was monitored for 10 days. As seen in Fig. 3a, untreated mice had a low survival rate, whereas the percentage of animals that survived after being treated with IP-administered rCD5L was 55%. Strikingly, IV treatment was much more efficient, yielding an extraordinary survival rate of over 70% (Fig. 3a).

To understand the molecular mechanisms involved in rCD5L-mediated protection, the progression of infection and immune responses were analyzed as depicted in Fig. 3b, for both IP or IV treatments. A group of animals was analyzed at 6 h post-CLP, after treatment with a single dose of rCD5L, injected 3 h post-CLP (Fig. 3b, top); another group receiving the two doses was analyzed 24 h after CLP (Fig. 3b, bottom). Bacterial counts were measured locally (peritoneal cavity) and systemically (blood, lungs, liver, kidneys), infiltrating leukocytes were counted in the peritoneal cavity, cytokines were

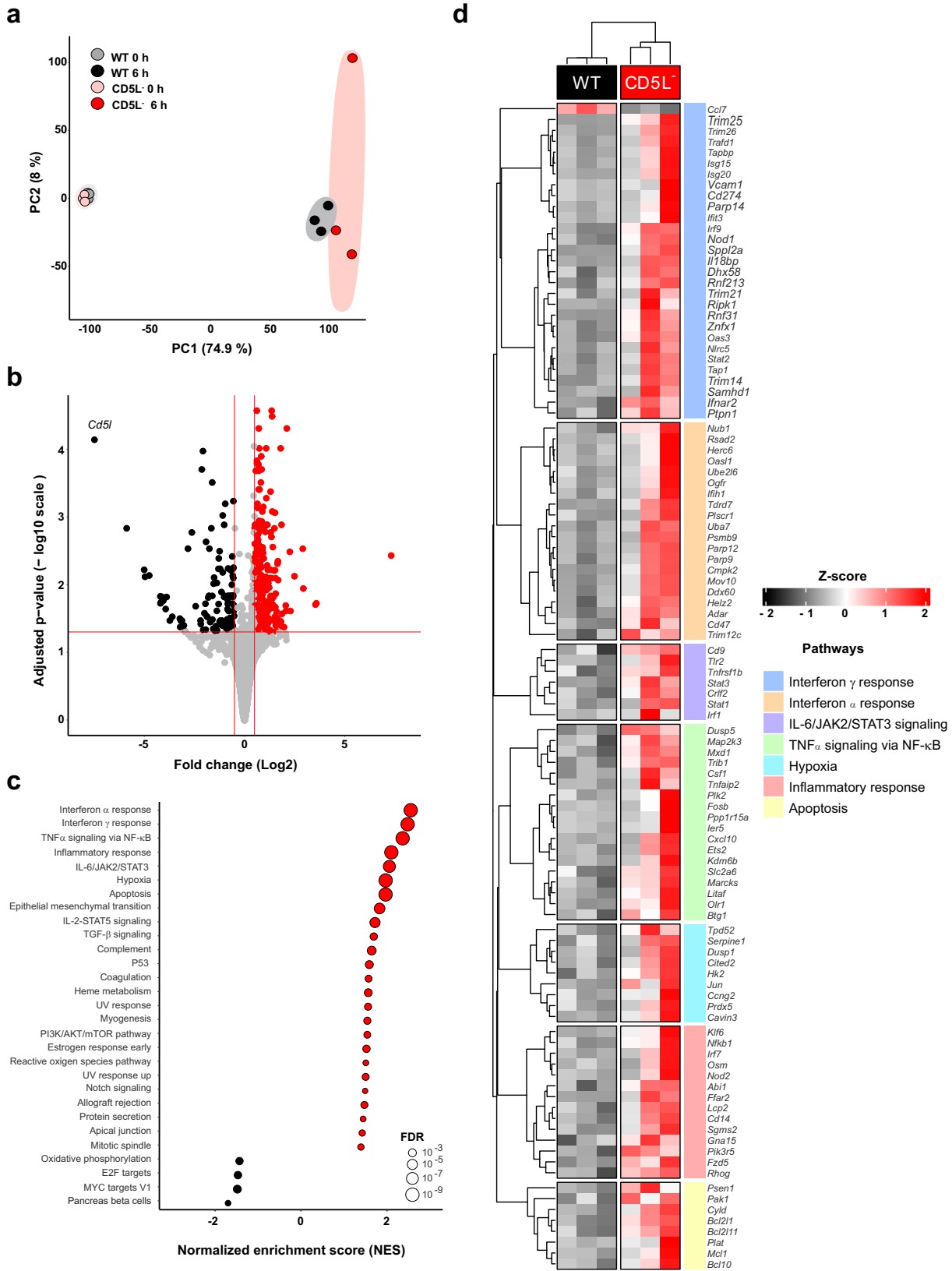

measured in the peritoneum and blood, and histopathology was performed in sections from lungs, liver and kidneys.

A global comparative analysis between the effect of rCD5L administered via IP vs. IV indicates that: i) IP and IV treatments led to significant increases in the total number of neutrophils in the peritoneal cavity 3 h after the first treatment, compared to untreated mice, but with greater amplitude in IV (-4.5-fold) than in IP treatment (-2.6-fold) (Fig. 3c). No other subsets showed major differences. ii) compared with the untreated group, IP injection resulted in a substantial reduction of some inflammatory cytokines, such as IL-6, TNF-α and IL-17, but only in the peritoneal cavity, whereas IV delivery of rCD5L caused a broad-spectrum reduction of inflammatory cytokines in both the

**Fig. 2 | RNA-seq analysis reveals important changes in inflammatory pathways in CD5L⁻ mice upon CLP, when compared with WT mice. a** Principal component analysis (PCA) of RNA-seq expression data for the 4 groups of peritoneal cells [WT and CD5L⁻ mice, either naive (0 h) or 6 h after CLP; *n* = 3]. **b** Volcano plot depicting differentially expressed genes in CD5L⁻ vs. WT mice, 6 h after CLP. Red dots represent genes expressed at higher levels in CD5L⁻ mice, while black dots represent genes with higher expression levels in WT controls. Grey dots represent genes bellow the cutoff of significant (|log₂ fold change| ≥ 0.5 and adjusted *P* values ≤ 0.05). Differential expression was evaluated using the Wald's test, followed by

adjustment for multiple testing with the Benjamini-Hochberg correction. Log₂-fold change values were shrunk with the apeglm method to increase the signal-over-noise ratio of the effect size. **c** Dot plot of gene set enrichment analysis (mouse hallmark gene set collection) for CD5L⁻ vs. WT mice, 6 h after CLP. The diameter of the dot indicates the degree of significance of the ontology term. Red dots represent terms enriched in CD5L⁻ mice, while black dots represent terms enriched in WT mice. **d** Heatmap of the relative expression values (z-score of each gene across samples) of the top 7 upregulated pathways. Only differentially expressed genes were represented, excluding genes belonging to more than one pathway.

peritoneum and blood (Fig. 3d, Fig. S7); iii) while IP treatment did not lead to significant differences in CFU counts from the different organs between untreated and rCD5L-treated mice, IV treatment was effective in reducing bacterial load in the lungs and liver (Fig. 3e). Also, the number of CFUs in the blood of IV-treated mice was reduced by > 200-fold, on average, 24 h after treatment, an effect not observed in IP-treated animals (Fig. 3e); iv) there were no major differences in the containment of organ damage between the two treatments, other than a very small decrease in lung histopathological score (Fig. S8a, b), and a reduction in serum AST levels in IV-treated, compared with untreated animals (Fig. S8c). A decrease in the absolute pathological score was, nevertheless, noted in some animals subjected to IV treatment. At the transcriptional level, aside from *Tnf-α* and *Il-1β* showing increased expression in the liver and kidney, respectively, 24 h after IV treatment, there were no significant differences in the levels of the main locally produced cytokines within the organs 6 to 24 h post-CLP (Fig. S9).

Overall, the hallmarks of the more effective treatment with intravenously administered rCD5L are rapid and massive recruitment of neutrophils, effective control of bacterial spread, and less systemic inflammation. Indeed, the administration of rCD5L directly into the circulatory system to combat high-grade CLP appears to better resemble the physiological response of WT mice to mid-grade CLP, as illustrated by a two-dimensional t-distributed stochastic neighbor embedding (t-SNE) representation of the high-dimensional data obtained in each treatment condition (untreated vs. IP-treated; untreated vs. IV-treated), overlaid with those obtained for the mid-grade CLP in CD5L⁻ mice (poor outcome) vs. WT mice (excellent outcome) (Fig. 4a). By linear discriminant analysis (LDA), we could determine that the parameters that maximize the separation of the IV-treated phenotype compared to the IP treatment are the increase in the number of recruited neutrophils and reductions in TNF-α and, to a lesser degree, IL-17, GM-CSF and IL-1α (Fig. 4b).

**Bio-distribution of bioactive CD5L in mid-grade CLP and therapeutic interventions**

An interesting observation arising from the transcriptional analysis of mid-grade CLP was that in peritoneal WT cells, *Cd5l* mRNA expression dropped dramatically after the surgery (Fig. 5a). By RT-qPCR analysis of peritoneal cells, we confirmed this same decrease at 6 h. However, extending the analysis to later time points allowed us to determine that 72 h after mid-grade CLP, *Cd5l* mRNA levels had recovered to baseline (Fig. 5b, left). A similar pattern was also observed in the liver (Fig. 5b, right). As CD5L protein levels are known to increase in infection sites, including the peritoneum during CLP[63,64], the discrepant mRNA levels might simply reflect the local presence or absence of CD5L-producing cells: the main source of CD5L, as well as the most abundant phagocytic cells in the peritoneum of naïve mice at steady state, are large peritoneal macrophages (LPMs), which rapidly disappear from the peritoneum after CLP or sham surgery; one to three days later, circulating monocytes infiltrate the peritoneum to repopulate this anatomical site[65,66]. Nevertheless, we measured, by ELISA, the levels of endogenous CD5L protein in the peritoneal cavity of WT mice upon moderate CLP aggression, and confirmed that peritoneal CD5L protein concentration increased sharply 6 h after mid-grade CLP, remaining

high during disease (Fig. 5c, left). Similar fluctuations, with lower magnitude, were observed in sham-operated mice.

On the other hand, blood CD5L levels dropped 6 h after mid-grade CLP and then recovered after 24 h (Fig. 5c, right). This observation made us consider the possibility of a mechanism of redistribution of CD5L from blood to peritoneum during the infection. To track circulating CD5L and screen for a possible direct transfer from blood to the peritoneum, we used CD5L⁻ mice where we can control the source of all CD5L present. We injected rCD5L IV into CD5L⁻ mice 3 h after mid-grade CLP and analyzed peritoneal fluid and blood 1 h later. The amount of rCD5L in the blood of the CD5L⁻ mice was quantified by ELISA at ~ 1-2 µg/ml and, importantly, became clearly detected in the peritoneal fluid (Fig. 5d). We found a statistically significant correlation between peritoneal and serum rCD5L in IV-injected CD5L⁻ mice, thus confirming our hypothesis of direct trafficking between blood and the infection site, and validating the intravenous administration of rCD5L as a useful therapeutic procedure (Fig. 5e).

Still, the quantification of absolute CD5L levels in our assays did not clarify whether the measured protein was in free state and therefore biologically active; or inert, if complexed with IgM. Using size-exclusion chromatography and western blotting, using conditions that prevented disrupting of intramolecular CD5L-IgM disulfide bonds[16,31], we confirmed that the ELISA assays indeed measured total CD5L (IgM-bound CD5L + free CD5L) (Fig. S10a–c), and that more than 90% of endogenous CD5L in the peritoneum and blood of healthy WT mice (0 h) is bound to IgM (Fig. 6a, Fig. S10d), as reported earlier[11]. During mid-grade CLP, however, there was a 6-fold increase in free endogenous CD5L in the peritoneal cavity, though not in blood, 24 h after surgery (Fig. 6a, Fig. S10d). Interestingly, IgM concentrations increased in the peritoneum while transiently decreased in the blood, in line with changes in total CD5L (Fig. 6b).

Next, we asked whether IV-injected rCD5L would reach the peritoneum of WT mice at sufficient levels, and whether the administered rCD5L remained as free, active protein, or would be readily integrated into IgM, as previously described[11,15]. We found that IV injection of rCD5L into WT mice resulted, 3 h later, in a significant increase in total CD5L levels in the peritoneal cavity and was equivalent to the IP treatment, decreasing to basal levels at 24 h (Fig. 6c). Furthermore, while the amount of total IgM-complexed CD5L did not noticeably change after IP or IV injection of rCD5L, the amount of free CD5L became significant, comparable to levels of the IgM-bound form, and was similar between IP or IV treatments (Fig. 6d, Fig. S10e). Collectively, these results suggest that the injected rCD5L retains its activity as a free protein, exhibiting uniform peritoneal distribution regardless of the administration route. Moreover, it appears to be consumed or excreted within 24 h of treatment.

**CD5L promotes neutrophil but not macrophage phagocytosis**

Having established that free bioactive CD5L is enriched within the peritoneal cavity during the physiological response of WT mice to mid-grade CLP, or following IP- or IV-therapeutic administration of rCD5L, we investigated the specific biological functions of the molecule that might play a role in combating the disease locally.

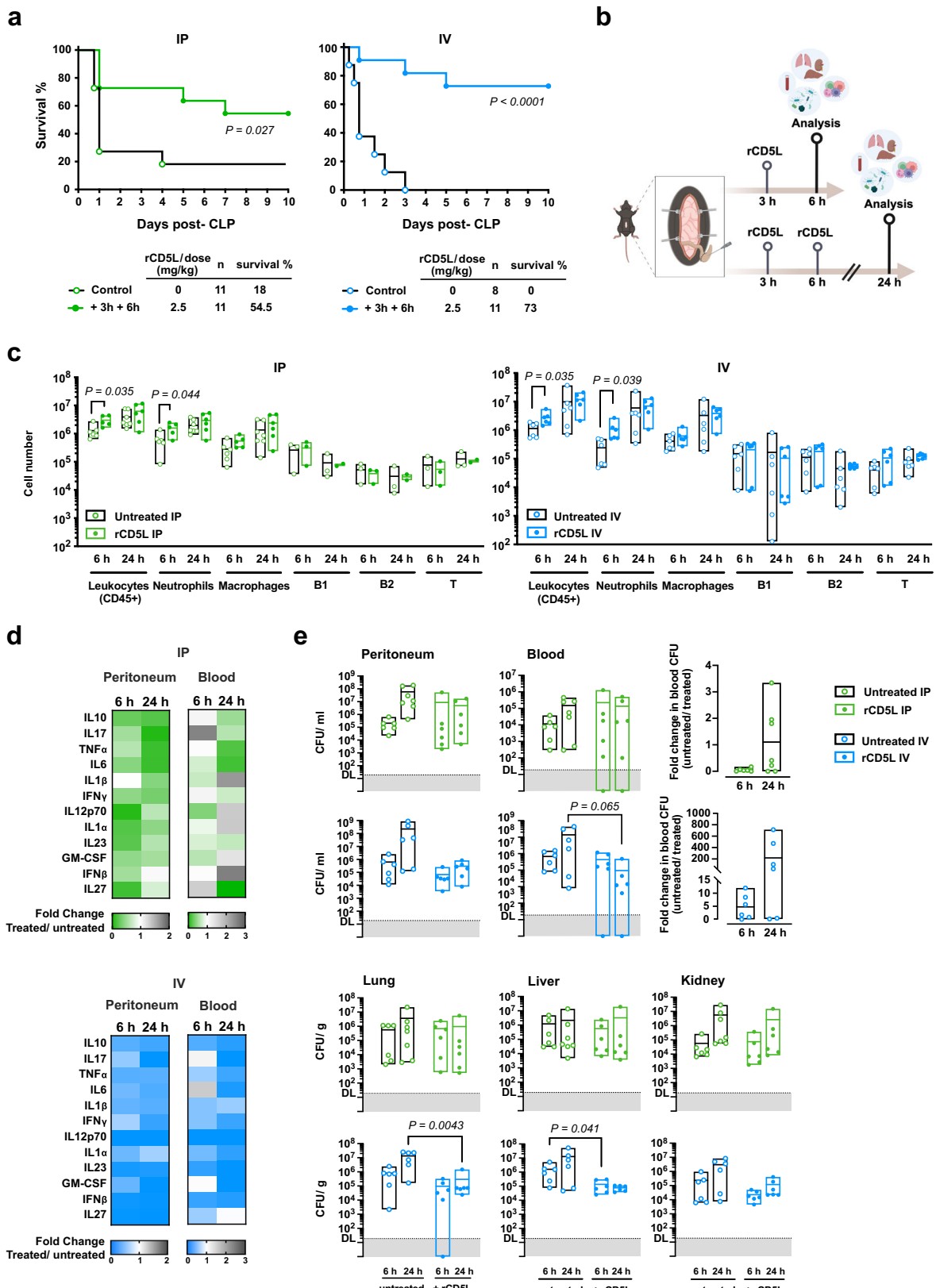

CD5L is described as binding to bacteria and inducing their aggregation[23]. However, there is no consensus that this induces phagocytosis or killing of the microorganisms[20,26]. We investigated whether rCD5L could contribute to an increase in phagocytosis by LPMs or neutrophils in vivo, by injecting pHrodo™ Red *E. coli* BioParticles™ together with 50 µg (2.5 mg/kg) of rCD5L, or with PBS only (0 µg

rCD5L), directly into the peritoneum of naïve WT mice. The pHrodo *E. coli* BioParticles become fluorescent in acidic pH solution, allowing to selectively identify bacteria that are inside the phagosomes[67]. Three hours after injection, peritoneal cells were recovered and analyzed by flow cytometry. Nearly all LPMs and about 15% of neutrophils had phagocytosed pHrodo *E. coli* Bioparticles, but these numbers were

**Fig. 3 | Administration of rCD5L reduces lethality of WT mice subjected to high-grade CLP. a** WT mice underwent CLP to induce severe disease. Mice were then injected with two doses of 2.5 mg/kg of rCD5L intraperitoneally (IP, left) or intravenously (IV, right) 3 and 6 h post-surgery. Kaplan–Meier curves compared survival between treated and untreated groups, with significance determined by log-rank (Mantel-Cox) test. Pooled data from 4 independent experiments, number of mice per group (*n*) is indicated. **b** Protocol for analysis after rCD5L treatment. Top - analysis at 6 h. Mice were injected IP or IV with 2.5 mg/kg rCD5L at 3 h post-CLP, or PBS, and euthanized 3 h later. Bottom - analysis at 24 h. Mice were injected IP or IV with 2 doses of rCD5L, 3 and 6 h after CLP, or PBS, and euthanized at 24 h. Image created with BioRender.com. **c** Flow cytometry-based absolute counts of cellular populations in the peritoneum of mice injected via IP or IV routes. Leukocytes, CD45$^+$; neutrophils, CD45$^+$CD11b$^+$Ly6G$^+$; macrophages, CD45$^+$CD11b$^+$CD11c$^-$F4/80$^+$; B1 cells, CD45$^+$CD19$^+$CD5$^+$; B2 cells, CD45$^+$CD19$^+$CD5$^-$; T cells, CD45$^+$CD3$^+$. Statistical

comparisons were made using two-tailed unpaired t-tests with Welch's correction. Mice per group: for IP-treatment groups, for Leukocytes, Neutrophils, and Macrophages at 6 h: 6 untreated and 5 IP-treated; at 24 h: 7 untreated and 6 IP-treated. For B1, B2, and T lymphocytes at 6 h: 3 untreated and 3 IP-treated; at 24 h: 3 untreated and 2 IP-treated. In all groups of IV-treatment, there were 6 mice each. **d** Fold change in indicated cytokines between IP- or IV-treated mice and PBS-injected controls. Cytokines quantified by bead-based multiplex immunoassays on samples from peritoneum and serum. **e** CFU counts in tissues from IP- or IV-treated mice and untreated groups. Statistical differences between groups were analyzed by two-tailed Mann-Whitney test. DL, detection limit = 20 CFU. Mice per group: for IP-treatment groups, at 6 h: 6 untreated and 6 IP-treated; at 24 h: 7 untreated and 6 IP-treated. For IV-treatment groups, 6 mice were in each group. Pooled data from at least 2 independent experiments (**c**–**e**). Floating bars show the minimum, average, and maximum values within each group.

## a

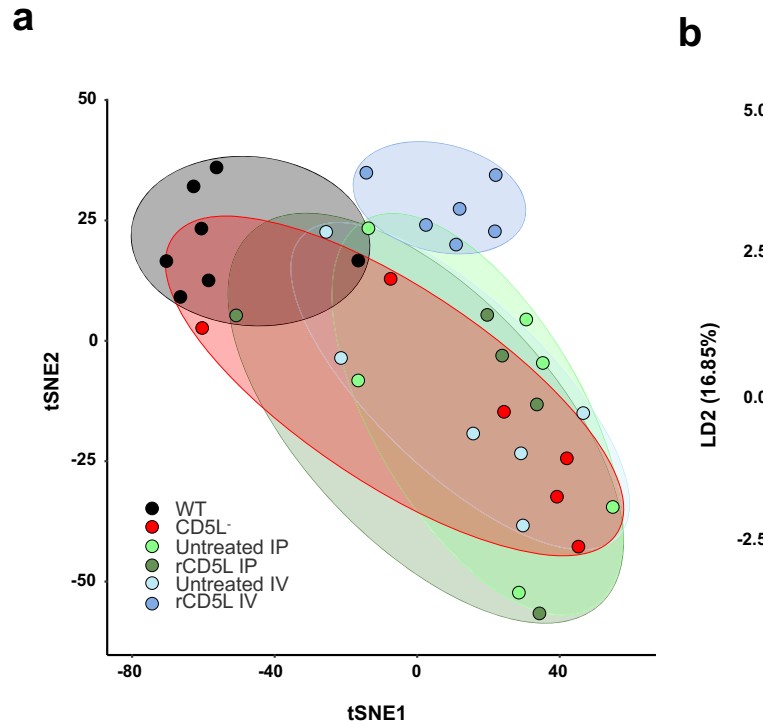

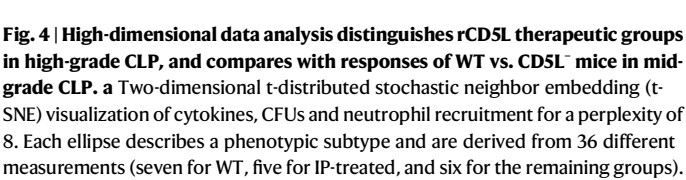

## b

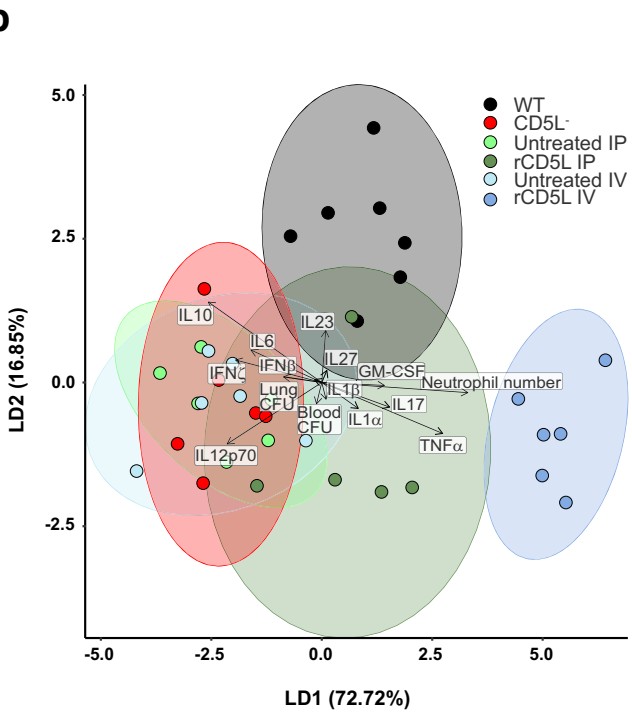

**Fig. 4 | High-dimensional data analysis distinguishes rCD5L therapeutic groups in high-grade CLP, and compares with responses of WT vs. CD5L⁻ mice in mid-grade CLP. a** Two-dimensional t-distributed stochastic neighbor embedding (t-SNE) visualization of cytokines, CFUs and neutrophil recruitment for a perplexity of 8. Each ellipse describes a phenotypic subtype and are derived from 36 different measurements (seven for WT, five for IP-treated, and six for the remaining groups).

Measurements were log$_2$ transformed and normalized by the mean value of the "untreated" or the CD5L⁻ subgroups for each setup [intravenous (IV); intraperitoneal (IP) or WT/CD5L⁻]. **b** Linear discriminant analysis (LDA) biplot showing the overall profile of treatment of CLP with rCD5L administered IV or IP, and mid-grade CLP in WT or CD5L⁻ mice. Ellipses show 95% confidence intervals for each treatment and vectors represent the contribution of each variable to the overall variance.

unchanged regardless of the presence of rCD5L (Fig. 7a). Based on the mean fluorescence intensity (MFI), indicative of the number of phagocytosed particles, we found no evidence that macrophages phagocytosed an increased number of pHrodo *E. coli* BioParticles following IP injection of rCD5L, compared with PBS controls (Fig. 7b, left panel). However, there was a slight indication that the presence of rCD5L might help neutrophils engulf greater numbers of particles (Fig. 7b, right panel).

To further evaluate the capacity, or inability, of rCD5L in helping macrophages phagocytosing bacteria, primary macrophages were recovered from the peritoneal cavity of WT mice 72 h after thioglycolate injection[68] and cultured with cecal bacteria, either naked (0 μg) or opsonized with rCD5L (2 or 10 μg) (Fig. 7c). Gentamicin-protection assays revealed, however, no contribution of

rCD5L for the phagocytosis of cecal bacteria (Fig. 7d). Finally, LPMs and thioglycolate-elicited neutrophils were recovered from the peritoneal cavity and placed in contact with pHrodo *E. coli* BioParticles, pre-coated, or not, with 10 μg rCD5L. The percentage of LPMs phagocytosing bioparticles increased over time, but was not influenced by the presence of rCD5L, nor was the number of bioparticles phagocytosed by these cells (Fig. 7e, f, upper panels), reinforcing the idea that CD5L does not contribute to macrophage phagocytosis. However, the number of rCD5L-coated pHrodo *E. coli* BioParticles engulfed by neutrophils in these assays was significantly greater than the number of naked bioparticles phagocytosed 6 – 24 h after the start of incubation, indeed suggesting a collaborative effect of the protein in neutrophil phagocytosis (Fig. 7e, f, lower panels).

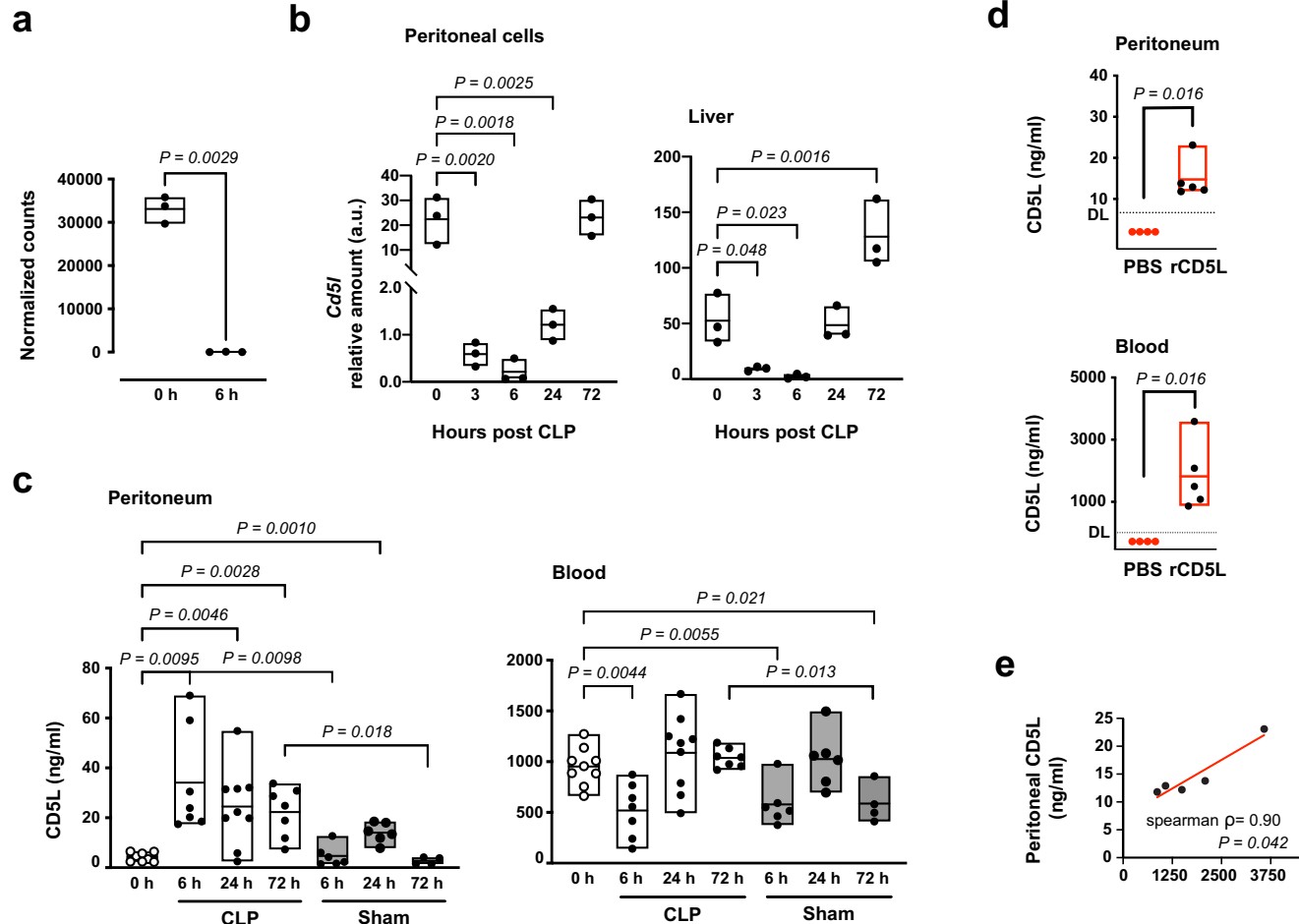

**Fig. 5 | Biodistribution of CD5L upon mid-grade CLP. a** Normalized counts of *Cd5l* transcripts obtained from RNA-seq analysis of peritoneal cells of WT mice at indicated times after CLP. Mice per group: 3. **b** RT-qPCR quantification of *Cd5l* expression, normalized with *Hprt1*, in peritoneal cells and liver tissue at indicated times after CLP. Mice per group: 3. **c** ELISA quantification of CD5L in peritoneum and blood of WT mice at indicated times after mid-grade CLP, or in sham operated animals. Mice per group: at 0 h: 8 in peritoneum, 9 in blood; CLP at 6, 24, and 72 h (peritoneum and blood): 7, 9, and 7; Sham at 6, 24, and 72 h (peritoneum and blood): 6, 6, and 4. **d** CD5L⁻ mice were subjected to mid-grade CLP and 3 h later injected IV with 2.5 mg/kg rCD5L, or PBS. Fluids from peritoneum and serum were

collected 1 h later and total CD5L was quantified by ELISA. DL, detection limit: 6.25 pg/ml. Mice per group: 4 for PBS; 5 for rCD5L. Pooled data from at least 2 independent experiments (**c**, **d**). **e** Scatterplot illustrates the correlation between peritoneal and blood rCD5L levels following IV injection, accompanied by Spearman's rank correlation coefficient and the two-tailed *P* value (*n* = 5). Statistical differences between groups analyzed by one-way ANOVA with Dunnett's multiple comparisons correction (**b**), two-tailed unpaired t-tests with Welch's correction (**c**), or two-tailed Mann-Whitney test (**d**). Floating bars show the minimum, average, and maximum values within each group.

## CD5L is a potent inhibitor of inflammation

As bona fide PRR, CD5L detects PAMPs and DAMPs, and the inactivation of these forms may decrease their associated inflammation[31]. During mid-grade CLP, we measured endotoxin levels and noticed that CD5L⁻ mice had substantially more LPS in circulation than WT mice (Fig. 8a). Also, there was a tendency to reduced LPS after IP- or IV-rCD5L administration in high-grade CLP (Fig. 8a). High mobility group box-1 (HMGB1) is a late mediator of endotoxin lethality in mice and has been highlighted as a prototypical DAMP involved in many inflammatory diseases[31,69,70]. We measured HMGB1 levels in different organs of WT and CD5L⁻ mice 72 h after mid-grade CLP, and whereas there were no differences in the lungs and kidneys, there was a substantial increase in HMGB1 in the liver of CD5L⁻ mice (Fig. 8b, c). Conversely, after IP- or IV administration of rCD5L in WT mice subjected to high-grade CLP, we detected a reduction of HMGB1 levels in the lungs and liver in the IP treatment, and in the liver and kidneys in the IV treatment (Fig. 8d, e). Altogether, this suggests that although rCD5L therapy generally improves the inflammatory profile of the

animals undergoing CLP, there are subtle nuances between therapeutic protocols that may affect target organs differently.

Next, we evaluated the potential of CD5L in controlling inflammation caused by in vivo administration of LPS, in a model conventionally termed sterile sepsis. WT and CD5L⁻ mice received a sublethal dose of 1.5 mg/kg LPS, and survival was monitored for 6 days. Although a small percentage of WT mice succumbed within 48 h after LPS challenge (18% lethality), the higher mortality of CD5L⁻ mice (73%) indicates that CD5L is critical for controlling the overt inflammation that leads to septic shock (Fig. 8f). In a parallel set of experiments, WT mice received a lethal dose (10 mg/kg) of LPS. Of these mice, 2 groups were treated with 2.5 or 5.0 mg/kg rCD5L, given IP 3 h after the LPS challenge, and a third group received PBS alone (0 mg/kg rCD5L). Strikingly, while the majority of untreated mice (93%) died within 2.5 days, 50% of mice that received rCD5L at 2.5 mg/kg were alive four days after LPS injection (Fig. 8g). Notably, those that received a dose of 5.0 mg/kg had a survival rate close to 80%. These experiments prove that CD5L has a genuine anti-

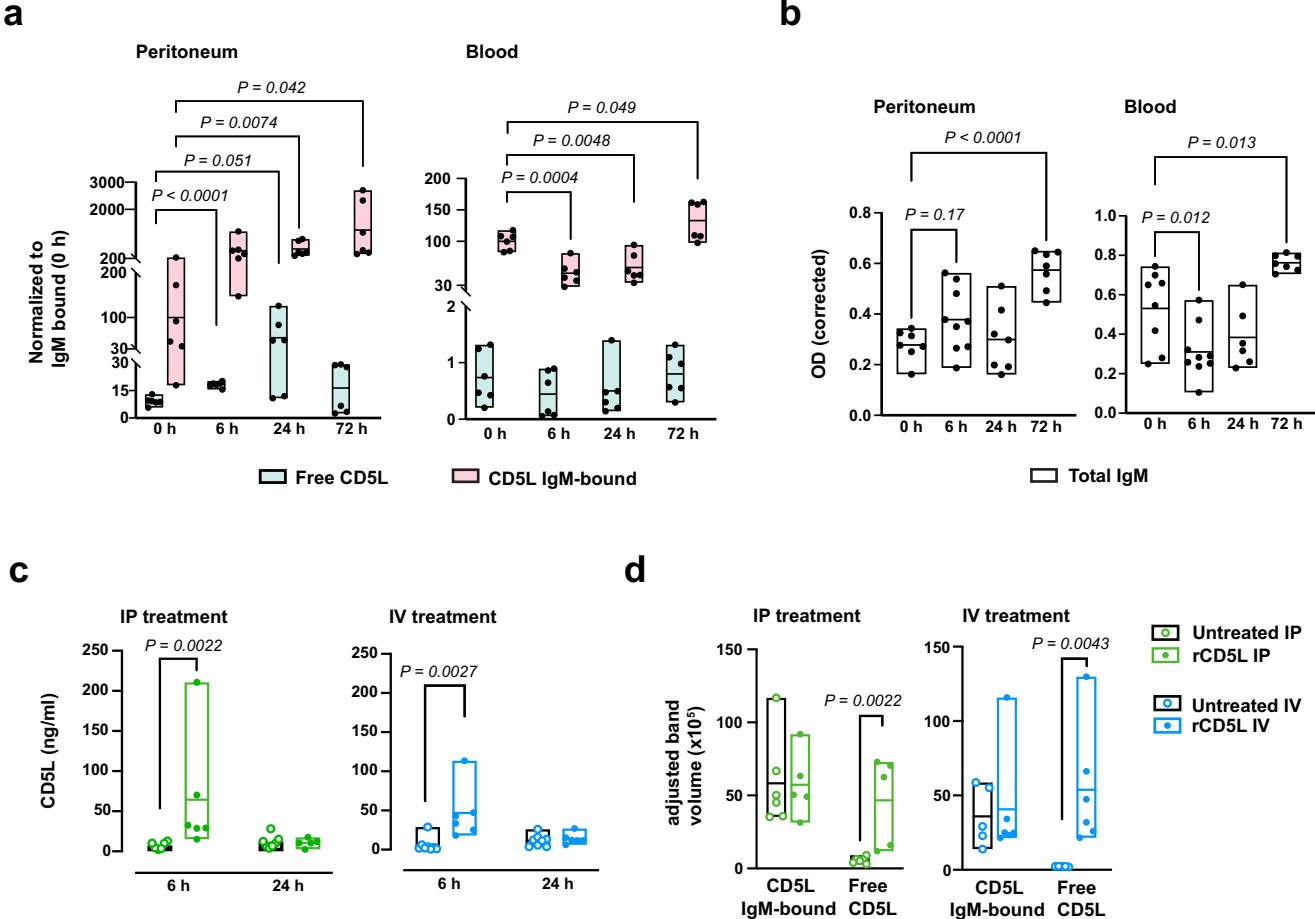

**Fig. 6 | CD5L bioactivity hinges on its free state dynamics. a** Determination of IgM-bound and free endogenous CD5L in peritoneal fluid and serum of WT mice at specified time points after CLP. Values were obtained through densitometric analysis of western blots normalized to baseline IgM-bound CD5L levels (100) in naïve mice. Each group comprised six mice. **b** ELISA quantification of total IgM in the peritoneum and serum of WT mice at specified times after CLP. Mice per group at 0, 6, 24, and 72 h were as follows: Peritoneum: 7, 9, 7, and 7; Blood: 8, 9, 6, and 7, respectively. **c** ELISA quantification of total CD5L in the peritoneal cavity of mice subjected to high-grade CLP, and IP- or IV-treated with rCD5L. At 6 h, mice had received one dose of rCD5L (2.5 mg/kg), or PBS, 3 h after CLP. At 24 h, mice had received 2 doses of rCD5L, or PBS, 3 and 6 h after CLP. Mice per group: IP treatment: 6 untreated and 6 treated, at 6 h; 7 untreated and 5 treated, at 24 h. IV treatment: 8

untreated and 6 treated, at 6 h; 8 untreated and 6 treated, at 24 h. **d** Determination of IgM-bound and free total (endogenous + recombinant) CD5L in the peritoneal cavity of WT mice treated with 2.5 mg/kg rCD5L, or PBS (untreated), via IP or IV routes, 3 h after high-grade CLP, and euthanized 3 h later. Values were obtained by densitometric analysis of western blot bands corresponding to IgM-bound and free CD5L. Mice per group: IP treatment: 6 untreated and 5 treated. IV treatment: 5 untreated and 6 treated. Pooled data from at least 2 independent experiments (**a**–**d**). Statistical differences between groups analyzed by two-tailed unpaired t-tests with Welch's correction (**a**), one-way ANOVA with Dunnett's multiple comparisons correction (**b**), or two-tailed Mann-Whitney test (**c**, **d**). Floating bars show the minimum, average, and maximum values within each group.

inflammatory action, extraordinarily effective even in the absence of a microbial challenge.

## IV administration of rCD5L orchestrates CXCL1-mediated neutrophil chemotaxis and activation

Our therapeutic protocols using rCD5L proved to be very successful in containing bacterial dissemination and treating acute infection caused by CLP, to which the effect of the protein on neutrophil recruitment undoubtedly contributed. In order to identify potential mediators of this effect, we analyzed inflammatory chemokines in the peritoneal cavity fluids and blood of WT and CD5L⁻ mice, retrieved 3 and 6 h after medium-grade CLP (Fig. 9a), and of WT mice subjected to lethal-grade CLP that were left untreated or were IV-treated with rCD5L (Fig. 9b). The inflammatory-chemokine arrays highlighted CXCL1, a chemokine connoted with neutrophil chemotaxis and activation[71], as the only consistently decreased in the peritoneal cavity and blood of CD5L⁻ mice, compared with WT controls, 6 h after mid-grade CLP (highlighted in Fig. 9c), and at the

same time significantly higher in the serum of rCD5L IV-treated mice, compared with the untreated group, 6 h after high-grade CLP (highlighted in Fig. 9d).

CD5L might therefore contribute to neutrophil recruitment and activation by regulating CXCL1 levels, as previously shown in a model of methicillin-resistant *Staphylococcus aureus*-induced pneumonia[26]. An alternative or complementary possibility is that CD5L activates neutrophils directly. To investigate these possibilities, we again turned to CD5L⁻ mice, in which we can manipulate the source of all CD5L present. In a scenario of mid-grade CLP, we administered 2.5 mg/kg of rCD5L intravenously and analyzed in the peritoneal cavity which cells were bound by rCD5L, likely targets of the protein. These included a small fraction of CD45-negative cells, T cells and neutrophils (Fig. 9e); however, the only population in which we detected by intracellular cytometry a clear increase in CXCL1 production after rCD5L binding were the CD45-negative cells (Fig. 9f). Of note, there were no changes in the frequency of CXCL1-producing cells within each cell subset in the time analyzed after rCD5L injection (Fig. S11).

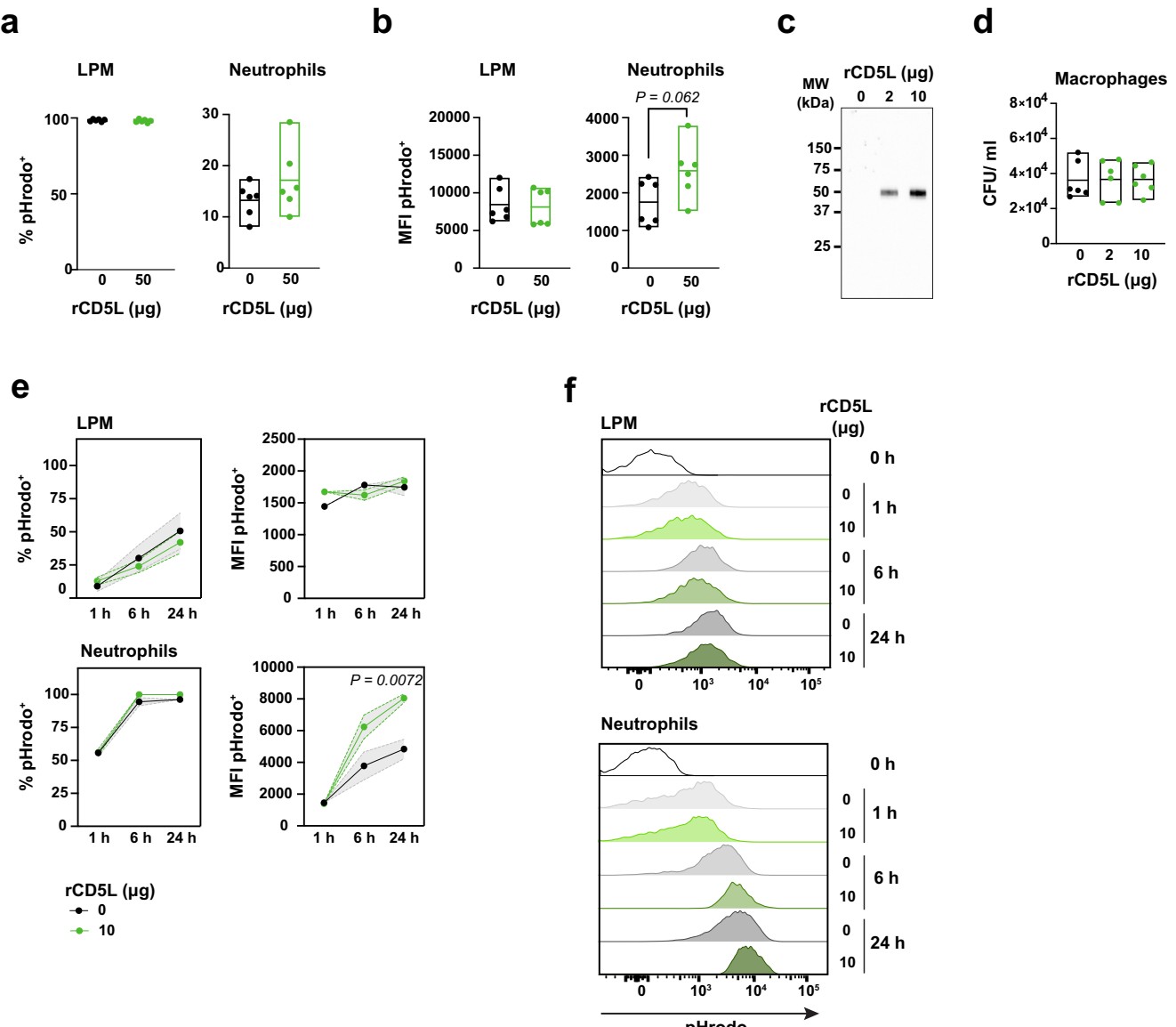

**Fig. 7 | CD5L promotes neutrophil phagocytosis. a, b** In vivo analysis of phagocytosis of pHrodo™ Red *E. coli* BioParticles™. WT mice were injected IP with 15 µl BioParticles and 50 µg of rCD5L, or PBS (0 µg). Peritoneal cells were recovered 3 h later and analyzed by flow cytometry. **a** Percentage of large peritoneal macrophages (LPMs, CD45⁺CD11b⁺/highF4/80high) or neutrophils (CD45⁺CD11b⁺Ly6G⁺) that internalized BioParticles. Mice per group: 6. **b** MFI (geometric mean) of pHrodo channel within the pHrodo-positive LPM or neutrophil populations. Statistical comparisons between groups determined by two-tailed unpaired t-tests with Welch's correction. Mice per group: 6. **c, d** Thioglycolate-elicited macrophages collected from mouse peritonea 72 h post-broth injection were infected with cecal bacteria pre-coated with 0, 2 or 10 µg of rCD5L. **c** After coating, a sample of cecal bacteria was boiled and subjected to SDS-PAGE. Detection of rCD5L bound to bacteria was performed via western blotting using primary anti-His tag mAb. **d** Gentamycin protection assay for phagocytosis analysis after 1-h incubation with pre-coated cecal bacteria at an MOI of 5 and enumeration of CFUs determined by serial dilution plating. Mice per group: 6. **a–d** Statistical comparisons between groups established by unpaired t-tests with Welch's correction. Pooled data from two independent experiments. Floating bars show minimum, average, and maximum values within each group. **e, f** Resident LPMs and thioglycolate-elicited neutrophils were collected from WT mouse peritonea 6 h post-broth injection and incubated with BioParticles pre-coated with 0 or 10 µg rCD5L for the indicated times before phagocytosis analysis by flow cytometry. **e** Percentage of LPMs (upper left panel) and neutrophils (lower left panel) that internalized BioParticles, and MFI values of pHrodo channel within the pHrodo-positive LPM (upper right panel) and neutrophil (lower right panel) populations. Statistical differences between groups analyzed by two-way ANOVA with Šídák's multiple comparisons test. Data shown are mean with SEM of *n* = 3 samples/group from one representative of two independent experiments. **f** Histogram overlays of representative examples of pHrodo channel fluorescence of LPMs and neutrophils incubated for the indicated times with uncoated (0) or rCD5L-coated (10) BioParticles.

The fact that, despite being targeted by rCD5L, neutrophils did not increase CXCL1 production suggested a different role for rCD5L on these cells. We thus analyzed the phenotype of neutrophils in WT mice after IP or IV administration of rCD5L, in high-grade CLP. We noted that besides a greater number of neutrophils being recruited to the peritoneum in IV-treated mice than in IP-treated mice, described above (Fig. 3c), there was also a rapid (6 h) increase in the expression of the activation marker CD11b on neutrophils from IV-treated mice compared with untreated controls (Fig. 9g). This neutrophil activation was not observed in IP-treated mice (Fig. 9g). Retrospectively, we analyzed CD11b levels in neutrophils from WT vs. CD5L⁻ mice in mid-grade CLP, and also confirmed the increased expression at 24 h of CD11b on WT neutrophils, compared with CD5L⁻ neutrophils (Fig. 9h). These observations suggest that a direct action of rCD5L is to induce neutrophils to acquire an activated phenotype, and that IV treatment may be more beneficial because

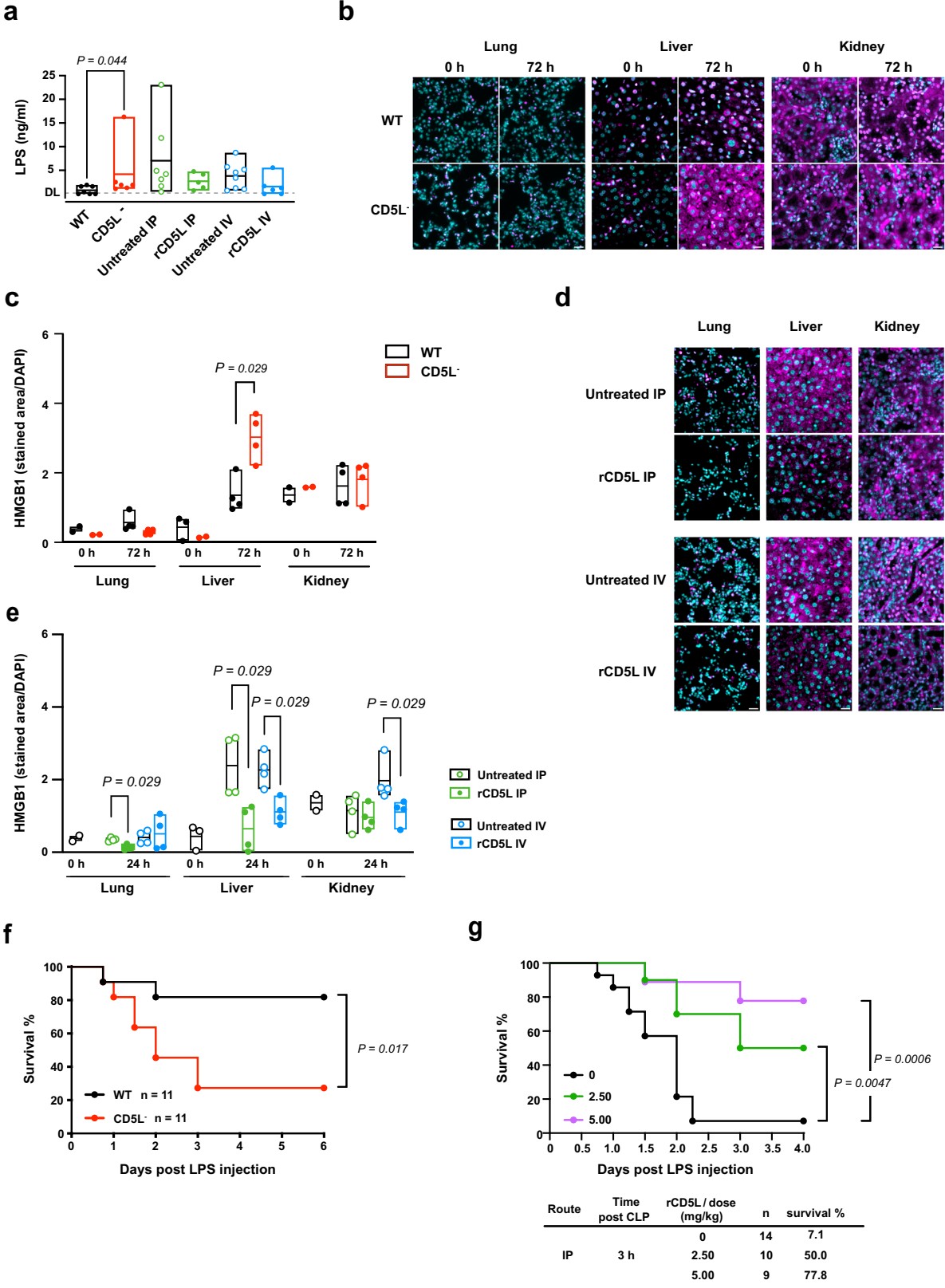

injection of the protein into the bloodstream activates circulating neutrophils more effectively than the intraperitoneal administration.

## Discussion

CD5L is a circulating glycoprotein that possesses significant immunological properties. Firstly, it functions as a pattern recognition receptor (PRR) with broad microbial recognition, making it effective in neutralizing various pathogens[72]. Secondly, it exhibits microbe-independent anti-inflammatory properties, efficiently containing inflammation[29,30]. Notably, in our study, the administration of rCD5L saved nearly 80% of mice from death induced by a lethal dose of LPS. Lastly, our findings indicate that CD5L plays a crucial role in neutrophil recruitment, activation and phagocytosis. The accumulated evidence

**Fig. 8 | CD5L reduces endotoxin and DAMP levels, and enhances mouse survival in septic shock. a** ELISA quantification of LPS in sera from WT and CD5L⁻ mice 72 h after mid-grade CLP; and in sera from WT mice subjected to high-grade CLP followed by rCD5L treatment. In these, mice were IP- or IV-injected with PBS (untreated), or with two doses of 2.5 mg/kg rCD5L, 3 and 6 h after surgery. Sera was collected and analyzed at 24 h. DL, detection limit: 0.16 ng/ml. The experimental groups comprised 7 WT and 6 CD5L⁻ mice for mid-grade CLP, and for high-grade CLP: 7 IP-untreated, 5 IP-treated, 8 IV-untreated, and 6 IV-treated mice. **b, c** HMGB1 protein expression in WT and CD5L⁻ organs, determined by immunofluorescence. **b** Representative images of HMGB1 expression in paraffin-preserved sections of lung, liver, and kidney of WT and CD5L⁻ mice. **c** Quantification of HMGB1-stained areas in 3 regions from 2 independent images, from 2 mice per CLP group (72 h), or from 1 naïve control mouse per group (0 h), normalized by the DAPI-stained area in the corresponding regions. **d, e** HMGB1 expression in organs of WT mice subjected to high-grade CLP, and IP- or IV-treated with two doses of rCD5L. **d** Representative images of HMGB1 expression in lung, liver, and kidney of treated and untreated mice. **e** Quantification of HMGB1-stained areas, as in (**c**). **a, c, e** Floating bars represent the minimum, average and maximum values within each group. Statistical differences between groups were analyzed using a two-tailed Mann-Whitney test. **b, d** Scale bar: 20 µm. **f** WT and CD5L⁻ mice were IP-injected with a sublethal LPS dose (1.5 mg/kg). Survival was monitored for 6 days. **g** WT mice were injected with a lethal LPS dose (10 mg/kg), and 3 h later with 2.5 or 5 mg/kg rCD5L, or left untreated (0). Survival was monitored for 4 days. **f, g** Kaplan–Meier curves were generated to compare survival between groups. Significance was determined by log-rank (Mantel-Cox) test. The graphical representation includes pooled data from 3 independent experiments for (**f**) and 2 independent experiments for **g**.

strongly suggests that CD5L may have a pivotal role in infection and is likely involved in sepsis.

Bulk transcriptomic analysis of peritoneal cells identified critically affected pathways in our CD5L⁻ mice that align with several aspects of the exacerbated responses observed in sepsis. Not surprisingly, the CD5L-deficient mice exhibited impaired immune and inflammatory responses during mid-grade CLP, rendering them susceptible to disease. Yet, the full extent to which the administration of rCD5L could counteract the adverse effects of high-grade CLP in WT mice had not been entirely anticipated. Rather than a mere transient increase in total CD5L as a consequence of therapy, it is the free form in which rCD5L enters the system that may elucidate its exceptional therapeutic efficacy in treating experimental sepsis. Typically, in a healthy state, the majority of CD5L is bound to serum IgM and remains biologically inactive. Conversely, under disease conditions, a fraction of CD5L uncouples from IgM, becoming a pleiotropic active factor. However, this process is slow and limited, with free CD5L reaching values not exceeding 15-20% of the total. In contrast, the therapeutic administration of rCD5L, whether IP or IV, led to the enrichment of free peritoneal CD5L, reaching levels comparable to those of the IgM-bound protein. This distinction in the form and enrichment levels of CD5L during therapeutic administration may contribute significantly to its outstanding therapeutic efficacy in experimental sepsis, suggesting promising avenues for exploration in human medicine.

With the therapeutic administration of rCD5L we obtained a very effective control of acute infection and inflammation and a superlative result in mouse survival in the CLP model. However, our results are at odds with a previous report describing that the IP administration of rCD5L, given at the time of CLP induction, actually aggravated mortality in septic mice[64]. The pathophysiology of sepsis provides a rationale to explain these apparently conflicting results. Sepsis arises from the disruption of the delicate balance between pro- and anti-inflammatory reactions triggered upon recognition of pathogen-associated molecular patterns (PAMPs) by specific receptors (e.g., TLRs)[73]. While both pro- and anti-inflammatory pathways are upregulated, the systemic release of cytokines and other mediators together with an increase of microbial by-products induces the massive activation of several cascades that lead to overt inflammation, progressive tissue damage, and multi-organ failure. If patients survive this stage, the increased apoptosis of immune cells together with cellular exhaustion contribute to immunoparalysis, characteristic of a late stage of sepsis, with increased susceptibility to secondary infections, namely by nosocomial opportunistic pathogens[74].

The main methodological difference between ours and the Gao et al. study[64] is the timing of rCD5L administration: they injected the recombinant protein at the time of inducing disease, whereas we included a delay of 3 h between CLP and the initiation of treatment. Given that CD5L is immunosuppressant, by being administered at the same time of disease initiation, rCD5L is most probably damping the natural beneficial response of the immune system during the initial inflammation, thus negatively affecting the clinical outcome of the disease. In our protocol, this initial phase is not affected, with rCD5L acting later to crush the uncontrolled systemic inflammation.

The contribution of CD5L in helping neutrophil phagocytosis that we describe was not wholly unexpected, despite some controversy on the role of the molecule in this process. However, we unveiled previously unsuspected functions of CD5L, which are to promote neutrophil recruitment and activation through the increase of circulating levels of neutrophil chemoattractant CXCL1. This chemokine was singled out in the inflammatory-chemokine array analysis of our two mouse models of CLP as the only chemokine that correlated with good disease outcome, and also consistent with its capacity to recruit and activate neutrophils.

CXCL1 is produced by several types of cell, including tissue macrophages[75], mast cells[76], neutrophils[77], and γδ T cells[78]. In particular, it has been shown that peritoneal macrophages and mast cells secrete CXCL1 in a TLR4-dependent manner in LPS- or zymosan-induced peritonitis, and these cells, together with specialized omental mesothelial cells which also produce CXCL1, coordinate the recruitment of neutrophils to capture peritoneal contaminants[76,79]. Using a mouse model of pneumonia-derived sepsis caused by *Streptococcus pneumoniae*, it was in fact shown that the CXCL1-mediated increase of neutrophil influx can be impactful to mouse survival by controlling bacterial growth and dissemination, and improving host survival[80]. Although the mechanism by which CD5L augments CXCL1 levels is not fully established, we show that it can target non-hematopoietic peritoneal cells to produce CXCL1, helping in neutrophil recruitment, and at the same time binds directly to neutrophils, likely inducing their activation. This fact may help explain why IV treatment is superior to the IP administration of rCD5L: injection of the protein into the blood might activate circulating neutrophils more effectively than the IP treatment and, together with CXCL1, increase neutrophil recruitment, leading to an enhanced and faster immune response, and ultimately to increased survival during acute infection and inflammation.

The role of neutrophils in sepsis is extremely important but paradoxical. On one hand, the inflammatory action of neutrophils is thought to contribute to tissue damage and the development of organ dysfunction during sepsis. On the other, it has been shown that mice undergoing sepsis have impaired activation and migration of neutrophils, resulting in failure of the host response to contain the infection locally, leading to the systemic spread of the pathogen[81]. In fact, one hallmark of sepsis is the excess of immature neutrophils in circulation, arising as consequence of defective emergency granulopoiesis together with fast turnover following severe infection, and that fail to migrate to the infection site in order to contain the microorganisms locally[82,83]. Altogether, neutrophil migration, antimicrobial activity, and the function of neutrophil extracellular traps (NETs) are impaired during sepsis, resulting in defective responses to a primary infection and potentially increasing the susceptibility to

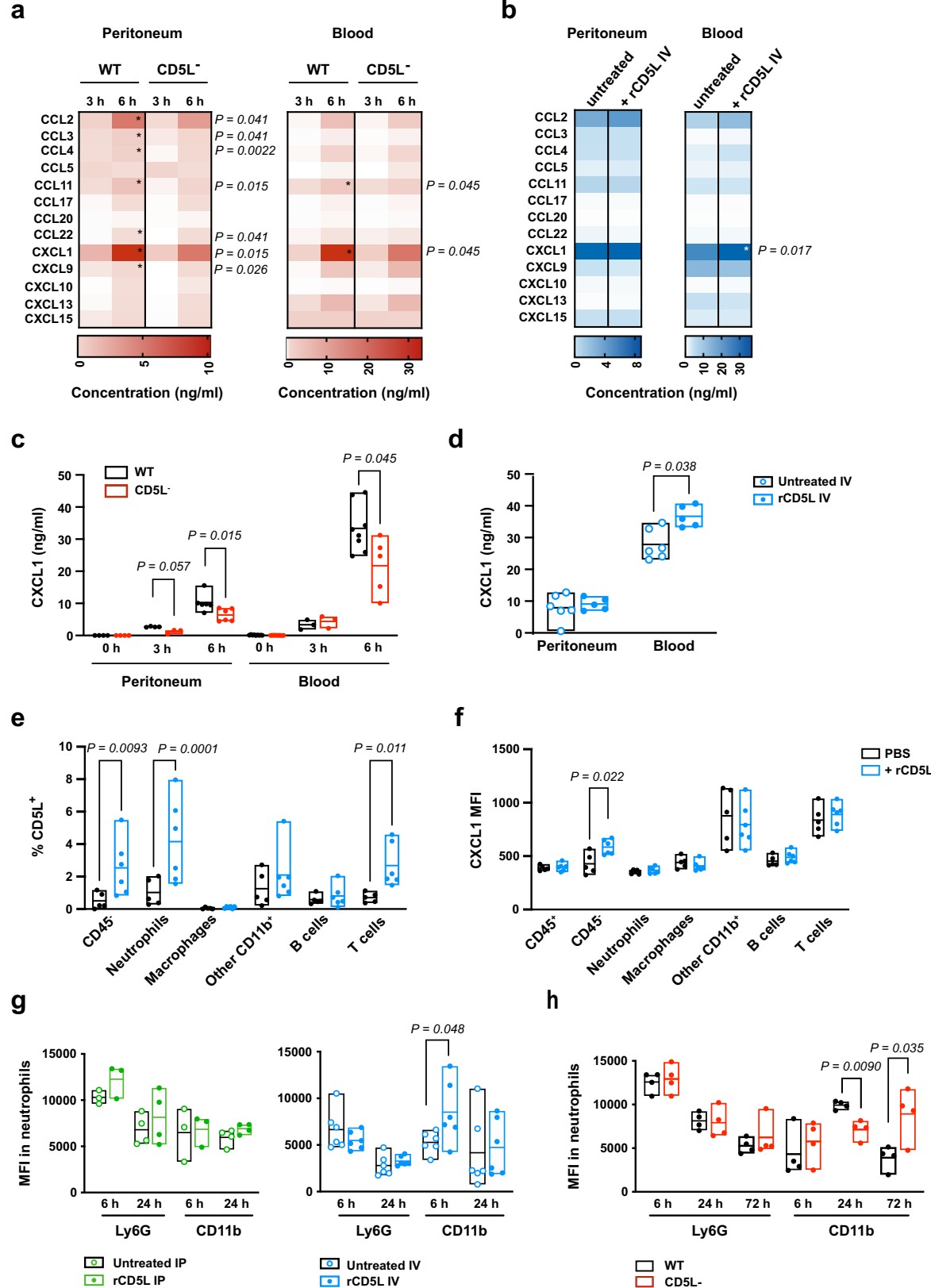

secondary insults[84]. The proposed therapeutic action of rCD5L likely works by leading to an increase in neutrophil activation and recruitment to the infection site, improving pathogen elimination locally and contributing to the reduction of the inflammatory phase, while diminishing the probability of occurrence of the late immune suppression stage (Fig. 10).

At present, treatment of human sepsis basically relies on infection control and extended life support, and it is noteworthy that no immunomodulatory strategy has so far proved efficient, despite strong scientific rationale and numerous interventional trials assessing relevant anti-inflammatory strategies[85]. The exogenous administration of rCD5L following the protocol we describe thus emerges as a

**Fig. 9 | CD5L promotes neutrophil chemotaxis via CXCL1, and neutrophil activation. a** Quantification of inflammatory chemokines in WT and CD5L⁻ mice at indicated time points after mid-grade CLP, using bead-based multiplex immunoassays. **b** WT mice subjected to high-grade CLP were IV-injected with 2.5 mg/kg of rCD5L, or PBS (untreated), 3 h later. Inflammatory chemokines were quantified 6 h post-CLP. **c** CXCL1 concentrations in WT and CD5L⁻ mice after mid-grade CLP (extracted from data presented in **a**). **d** CXCL1 concentrations in untreated and IV-treated WT mice after high-grade CLP (extracted from data presented in **b**). **a–d** Pooled data from at least 2 independent experiments, analyzed by two-tailed Mann-Whitney test. **a, c** Mice per group: Peritoneum: 4 WT and 4 CD5L⁻ at 0 h, 4 WT, and 3 CD5L⁻ at 3 h, 6 WT, and 6 CD5L⁻ at 6 h; Blood, 9 WT and 9 CD5L⁻ at 0 h, 3 WT, and 3 CD5L⁻ at 3 h, 8 WT and 5 CD5L⁻ at 6 h. **b, d** Mice per group: 6 in untreated, 5 in rCD5L IV-treated. **e** CD5L⁻ mice were injected IV with rCD5L (2.5 mg/kg) 3 h after

mid-grade CLP and peritoneal cells were recovered 3 h later. Percentage of CD5L-bound cells within CD45⁻ cells; neutrophils (CD45⁺CD11b⁺Ly6G⁺), macrophages (CD45⁺CD11b⁺F4/80⁺), other CD11b⁺ (CD45⁺CD11b⁺F4/80⁻Ly6G⁻), B cells (CD45⁺B220⁺), and T cells (CD45⁺CD3⁺). **f** MFI values of the CXCL1 channel within CXCL1⁺ cells in each subset defined in **e**. **e, f** Pooled data from 2 independent experiments, analyzed by two-way ANOVA. Mice per group: 5 untreated and 6 rCD5L-treated. Ly6G and CD11b MFI in neutrophils collected from the peritoneal cavity of rCD5L IP- (left panel) or IV-treated (right panel) WT mice, after high-grade CLP (**g**), or WT vs. CD5L⁻ mice after mid-grade CLP (**h**). Mice per group: IP treatment: 3 at 6 h, 4 at 24 h; IV treatment: 6 at 6 h, 6 at 24 h (**g**); IP and IV treatment: 4 (**h**). Pooled data from 2 independent experiments, statistical comparisons between groups were established by two-tailed unpaired t-tests with Welch's correction.

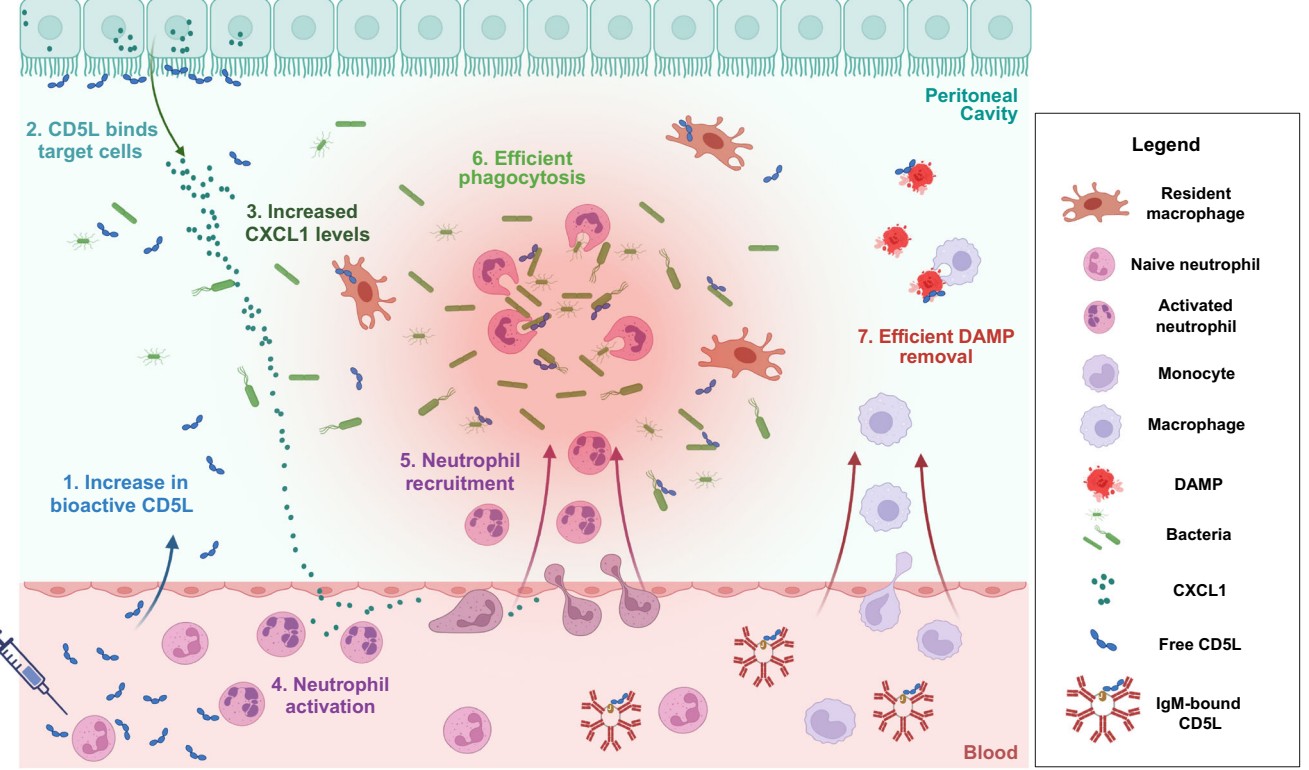

**Fig. 10 | Mechanistic action of rCD5L.** Intravenous administration of rCD5L leads to elevated levels of free bioactive protein in the peritoneal cavity (1). Upon binding to target cells of non-hematopoietic origin (2), rCD5L induces significant production of the CXCL1 chemokine (3). The production of CXCL1 establishes a chemotactic gradient, resulting in heightened neutrophil activation (4) and recruitment from the bloodstream (5). The exact mechanism behind neutrophil activation, whether through increased CXCL1 levels or direct action of rCD5L, remains unclear. Increased neutrophil presence in the peritoneal cavity, coupled with the synergistic effect of rCD5L in bacterial binding and enhanced phagocytosis, facilitates efficient bacterial clearance (6). rCD5L aids in the effective removal of DAMPs, likely from both the bloodstream and the peritoneum (7). Endogenous local release of CD5L from resident macrophages contributes to the overall pool of free CD5L at the site of infection. This illustration is for explanatory purposes and is not drawn to scale. Created with Biorender.com.

potential therapeutic alternative for the treatment of acute infectious and inflammatory diseases. The therapeutic potential of CD5L relies on a unique combination of biological properties that can be adjusted to obtain a controlled clinical benefit: basically, our therapy induces a rapid increase of the concentration of free active CD5L to obtain a customized response to the infectious challenge. We trust that the noticeable benefits obtained in our animal models can quickly be exploited and translated to clinical medicine.

## Methods
### Ethics statement
All experiments were conducted in 8- to 12-week-old male and female C57BL/6 J mice purchased from Charles River and bred in-house following Portuguese (Portaria 1005/92) and European (Directive 2010/

63/EU) legislations concerning housing, husbandry and welfare. The project was reviewed and approved by the Ethics Committee of the Instituto de Investigação e Inovação em Saúde (i3S), Universidade do Porto, and by the Portuguese National Entity Direção Geral de Alimentação e Veterinária (license reference: 009951).

### Development of *Cd5l⁻/⁻* mice and immunophenotyping
*Cd5l⁻/⁻* mice were generated using CRISPR/Cas9 engineering, as described previously[86]. In brief, an sgRNA (10 ng/µl) targeting the 5'-GGCTGATATGATGCGCCACG-3' sequence located in exon 3 of the *Cd5l* gene, Cas9 mRNA (20 ng/µl), and replacement ssDNA oligonucleotide (10 ng/µl, from IDT) containing the 60-nucleotide sequences from each side of the deletion, separated by an *Eco*RI restriction site and three tandem stop codons, termed ssDNA1 (Supplementary data 1), in

10 mM Tris-HCl pH 7.5, 0.1 mM EDTA, were introduced by pronuclear injection of C57BL/6 J fertilized oocytes, using standard procedures[87]. sgRNAs were generated by annealing oligos sgRNA1 and sgRNA2, and introducing them into plasmid gRNA basic[86]. Plasmids were linearized and sgRNAs synthesized by in vitro transcription (Megashort Script in vitro T7, Ambion), purified (Megaclear RNA kit, Ambion), and analyzed on Experion (Bio-Rad) for integrity, purity, and yield. Gene edition was assessed by PCR from tail genomic DNA and confirmed by direct sequencing.

CD5L[-/-] were crossed with C57BL/6 J WT mice and the obtained CD5L [+/-] progeny couples were bred to obtain CD5L [-/-] and CD5L [+/+] progeny. From this point, CD5L [+/+] and CD5L [-/-] pups were grouped and housed together conforming to sex and age to be used for the intended procedures and to establish new breeding couples according to the genotype. CD5L [-/-] were crossed regularly with C57BL/6 J WT to generate new CD5L [+/-] couples to be bred together.

Female mice were used in therapeutic experiments, due to their significantly lower weight compared to males, thus requiring a smaller amount of rCD5L to be administered. There have been mixed reports on how sex affects sepsis in clinical and rodent sepsis models, with most reporting improved survival in females[88,89]. We did not find a gender bias effect in our experimental conditions, in line with a recent report that included a large cohort of mice[90], therefore both males and females were used in our studies. Whenever possible, mice were assigned to specific groups randomly.

In-depth immune-phenotyping analysis was performed on spleen, thymus, peripheral blood and peritoneal cavity of 12-week-old WT and CD5L[-] mice (3 males, 2 females per group) by multi-parametric flow cytometry. Leukocytes were isolated from thymi and spleens by combined mechanical treatment and enzymatic digestion using GentleMACS™ octo dissociator system (Miltenyi). Cell suspensions were numerated for all samples on AttuneNxT flow cytometer. Erythrocytes were lyzed using RBC Lysis Buffer (eBioscience, #00-4333-57). Fc-receptors were blocked using anti-CD16/CD32 (#24G2). Cells were stained in one step with the corresponding antibody panel (Supplementary data 2, 3), washed and analyzed within 4 h after organ collection. Dead cells were removed using Live/Dead SytoxBlue staining (Thermo, #S11348). Datasets were acquired on a 5-laser BD LSRFortessa™ SORP cell analyzer using an HTS plate reader on a standardized analysis matrix using BD FACSDiva 8.0.1 software. Data were visualized using radar plot on Excel or Tableau Desktop software. Frequencies of cell subsets on all mice of the study were pooled in one spreadsheet per panel per organ, transformed in asinh, and centered to the mean. The resulting variable list was run in the SIMCA® Multivariate Data Analysis software. First, distribution of samples in PCA score was considered to remove potential outliers. Next, an OPLS-DA method was applied on the remaining dataset to identify groups of samples presenting similar types of variations. Overall, the distribution of mice was similar before and after OPLS-DA modelling for each experimental group. Variables important for this projection (VIP) were selected with a VIPpred value over 1. A final PCA was run on data only coming from these selected predictive variables to ensure that the OPLS-DA model was not overfitting the dataset. List of phenotypes with VIP > 1 were further visualized on moustache plots using Tableau Desktop software.

## Mouse recombinant CD5L
Mouse rCD5L protein was obtained from INVIGATE GmbH. The recombinant protein was produced in mammalian HEK293T cells, grown in roller bottle cultures. Purification procedures included ultrafiltration (5 kDa Sartocon Slice Cassette), and metal affinity chromatography (IMAC) using Ni-NTA-Agarose-Resin (Genaxxon bioscience). For further purification, samples were passed through Q-Sepharose FF (GE Healthcare). Endotoxicity in the rCD5L solution

was 0.21 EU/µg protein, determined by *Limulus* amebocyte lysate chromogenic endpoint assay (Associates of Cape Cod Europe GmbH).

## Immunofluorescence
Mouse peritoneal cells were adhered to poly-L-lysine-coated glass slides for 30 min at 37 °C followed by 4% PFA fixation. Surface staining with anti-F4/80 mAb was performed followed by AlexaFluor 594-coupled anti-rat secondary antibody (Supplementary data 4). Intracellular staining with goat anti-CD5L polyclonal antibody followed by AlexaFluor 488-coupled donkey anti-goat secondary antibody after permeabilization with Triton X-100 (Sigma). 4′,6-diamidino-2-phenylindole (DAPI) was used as nuclear counterstaining. Cells were visualized in a Leica SP5 confocal microscope.

For HMGB1 staining, lung, liver and kidney tissue samples were fixed in neutral buffered 10% formalin and paraffin embedded. Histologic sections of 4 µm thickness were cut with HM335 paraffin microtome (Microm) and mounted onto SuperFrost® Plus glass slides (Thermo Scientific). Tissue sections were deparaffinized in xylene for 10 min, followed by hydration through sequential ethanol gradient (100%, 96%, 70%). Antigen retrieval was performed in 0.05 M Tris-Buffered Saline, 0.1% Tween-20 (TBS-T) pH 9, and heated at 95 °C for 30 min. Staining was done using UltraVision Quanto Detection System HRP DAB (Epredia), with some modifications. Endogenous peroxidases were blocked for 10 min with hydrogen peroxide and the kit's protein block was replaced with 10% mouse-on-mouse IgG blocking reagent diluted in TBS-T for 1 h, followed by 10% goat serum solution diluted in TBS-T for 1 h at RT.

Tissues were incubated with 10 µg/ml anti-HMGB1 (Supplementary data 4), diluted in Ab Diluent (Dako), overnight at 4 °C. After extensive TBS-T washing, the primary antibody amplifier was incubated for 10 min before washing again with TBS-T. A secondary antibody, 4 µg/ml AlexaFluor 568-coupled goat anti-mouse IgG, diluted in Ab Diluent, was added and incubated for 1 h, followed by TBS-T and deionized water rinses. Nuclear counterstain was performed with DAPI for 5 min. Slides were mounted using anti-fade mounting medium (Thermo Fisher Scientific) and visualized using a Leica DMI6000 FFW microscope and acquired in LAS X software.

The HMGB1 stained area was quantified in Fiji (ImageJ2 V2.3.0), using the "Threshold" tool for image segmentation. In each image, three representative regions of interest (ROIs) were defined for quantification. Consistent threshold values were applied to each organ in each group to ensure uniform identification of HMGB1. Individual quantification for each channel was performed within these refined regions, using the "measure" function to determine the percentage of area occupied by HMGB1 and DAPI relative to that of total cells. The HMGB1 area was then normalized by the DAPI area to account for differences in cellular number and distribution. Analysis was performed in two images per organ from 2 different animals per group and time point.

## ELISA
Mouse CD5L quantification was performed using Mouse CD5L ELISA Pair Set (SinoBiological), mouse CXCL1 was quantified using CXCL1/KC DuoSet ELISA (R&D Systems), LPS using Mouse LPS ELISA Kit (Biorbyt Ltd.), AST using mouse AST ELISA Kit (Elabscience), and creatinine using Cr ELISA kit (MyBioSource.com).

Total IgM was quantified by sandwich ELISA. Briefly, 96-well plates (Nunc Maxisorp) were coated with 5 µg/ml goat anti-mouse Ig overnight at 4 °C. The plate was blocked with 2% BSA in Tris-buffered saline pH 7.4, 0.05% Tween-20 (TBS-T), for 1 h at RT. After washing with TBS-T, HRP-labeled goat anti-mouse IgM was added and left to interact for 2 h at RT. After a last TBS-T washing, TMB substrate (Biolegend) was added and color development was stopped with 2 N $H_2SO_4$. Optical density at 450 and 570 nm was recorded (Synergy 2, BioTek). Optical density values from samples obtained at 570 nm were subtracted from

                     

those at 450 nm and background values from blank wells were subtracted from samples.

## RNA extraction, bulk RNA sequencing, and RT-qPCR

RNA extraction was performed in cells and tissues using the PureLink™ RNA Mini Kit (Invitrogen) with on-column DNAse (Roche) treatment. Quantification and purity assessment were analyzed in NanoDrop 1000 spectrophotometer or in Agilent 2100 Bioanalyzer system. RNA integrity numbers (RIN) above 6.5 were used for library preparation and reverse transcription followed by sequencing using Illumina PE150 technology (Novogene).

RT-qPCR was performed in RNA samples reverse transcribed using NZY Reverse Transcriptase (NZYtech). Quantitative analysis of gene expression was performed using real-time PCR (CFX96, BioRad) with 2x iTaq Universal SYBR Green Supermix (BioRad). The relative expression levels of each gene of interest were calculated using the $2^{-\Delta\Delta CT}$ method, using *Hprt1* as reference. Primers are listed in Supplementary Data 1. *CdSl* relative expression was measured from PBS-perfused organs to eliminate possible contamination from blood.

## Bulk RNA-seq data processing and analysis

RNA-seq reads were pre-processed in TrimGalore prior to alignment to remove low-quality and adapter sequences. Sequence alignment was carried in STAR v2.7.10a with the mouse GRCm39 reference using Gencode vM33 gene models. Counts per gene were generated by STAR. PCA was performed using the rlog-normalized count data to inform about sample variability and structure across conditions.

Differential expression between CD5L⁻ and WT conditions ($n = 3$ each) was assessed in DESeq2 (v.1.34). Log$_2$ fold change results were shrunk using the *apeglm* package to remove noise (DE genes with low counts and/or high dispersion values) while preserving significant differences.

Pathway enrichment analysis was performed in fGSEA using a pre-ranked list of genes calculated using the signed log$_2$ fold change times the -log$_{10}$ of the $P$ value. Enrichment was calculated for the mouse hallmark gene sets that were retrieved from the Molecular Signatures Database (v2023.2.Mm).

Heatmap of differentially expressed genes between CD5L⁻ and WT conditions for the top 7 up-regulated hallmark pathways was plotted using ComplexHeatmap package.

## Histopathology

Tissues were fixed in 10%-buffered formalin and paraffin-embedded. Serial consecutive 2 μm-sections were cut and processed for routine staining (hematoxylin and eosin). Slides were observed through light microscopy (Nikon Eclipse 50i) and evaluation was conducted by a pathologist blinded to the experimental conditions. Lung, liver, and kidney sections were examined and histopathological findings were scored from 0 to 4, according to a four-level detailed and previously validated severity scoring system[91], where "0 or absent" was assigned in the absence of alterations or in conditions similar to the control; "1 or minimal" corresponded to histological changes that were barely noticeable or considered minor, small, or infrequent warranting no more than the least assignable grade (0-10%); "2 or mild" corresponded to noticeable but not prominent histologic changes. For focal, multi-focal, or diffusely distributed lesions, this grade was used for processes involving 11-20% of the tissue examined; "3 or moderate", when histological changes were prominent and their distribution affected 21-40% of the tissue examined; "4 or severe", when histological changes were an overwhelming feature and affected 41-100% of the tissue examined.

## Flow cytometry and multiplex assays

Cells were recovered and $1 \times 10^6$ cells were washed with PBS before staining with a viability dye (Fixable viability dye, eBioscience). After,

cells were kept in flow staining media (FSM, PBS containing 2% FBS) and Fc receptors were blocked (TruStain FcX anti-mouse CD16/32 antibody) before staining with fluorochrome-conjugated antibodies against surface markers or anti-CD5L unconjugated antibody followed by secondary antibody (Supplementary data 4).

In assays detecting CXCL1 by flow cytometry, collected peritoneal cells were incubated 4 h with 5 μg/ml brefeldin A (Sigma-Aldrich) before surface staining with appropriate markers, as above. After, cells were fixed and permeabilized using FoxP3 Transcription factor staining buffer set (eBioscience). Intracellular staining with unconjugated anti-CXCL1 antibody was performed after permeabilization, followed by incubation with a secondary anti-rabbit antibody. Cells were analyzed in FACSCanto II or LSRFortessa flow cytometers (BD), using FACS Diva software.

Mouse cytokine (LEGENDplex™ Mouse Inflammation Panel, BioLegend) and chemokine (LEGENDplex™ Mouse Proinflammatory Chemokine Panel, BioLegend) were quantified in mouse sera or peritoneal fluid. Samples were acquired in an Accuri C6 (BD) flow cytometer.

All cytometry data were analyzed in FlowJo software (V10.10.0).

**SDS-PAGE and western blotting.** Samples were mixed with 6x Laemmli sample buffer containing or not DTT, denatured for 10 min at 95 °C and loaded on 7.5 % (for CD5L detection in IgM-associated or free form) or 10% SDS-polyacrylamide gels (SDS-PAGE) and separated for 45 min at 200 V (Bio-Rad). Samples were transferred to nitrocellulose membranes (TransBlot Turbo, Bio-Rad), blocked with 5 % non-fat milk in TBS-T for 1 h at RT, and incubated with primary antibody in 3% non-fat milk in TBS-T, overnight at 4 °C. Membranes were washed three times with TBS-T for 10 min and incubated with HRP-conjugated secondary antibody in 3% non-fat milk in TBS-T for 1 h. ECL select (Amersham) was used to develop chemiluminescent signals and images were acquired in ChemiDoc system (BioRad).

**Size Exclusion Chromatography.** WT mouse serum (200 μl) was loaded onto a Superose 12 10/300 GL (Cytiva) column for size exclusion chromatography (AKTA Purifier 10 FPLC System, Cytiva). FPLC was performed at 0.2 ml/min with PBS, and 0.5 ml fractions were collected. 10 μl of each fraction were run in SDS-PAGE, and detected for CD5L and IgM by western blotting. ELISA quantification of CD5L was performed in parallel.

**Phagocytosis assays.** Preparation of rCD5L-coated pHrodo particles. 2 mg of particles (pHrodo™ Red *E. coli* BioParticles™ Conjugate for Phagocytosis, Thermo Fisher) were reconstituted with 500 μl TBS with 5 mM CaCl$_2$ and sonicated for 5 min. Particles were split into 2 tubes, each containing 250 μl of reconstituted particles: 10 μg rCD5L were added to one tube and the other was left without protein, as control. Protein was incubated with particles for 1 h at 4 °C with rotation, followed by washing 3 times with PBS and resuspending each sample in 250 μl PBS.

Peritoneal cell recovery for in vitro assays. Neutrophils and macrophages were recovered from the peritoneal cavity respectively at 6 or 72 h after injection of 2.5 ml of 3% thioglycolate IP. For that, mice were euthanized, and 5 ml RPMI (HyClone GE Healthcare Life Sciences) with 5% FBS were injected in the peritoneal cavity and left for 1 min before recovery by aspiration with a syringe. Macrophages were further enriched by panning for 30 min, and neutrophil enrichment was confirmed by flow cytometry.

In vitro phagocytosis assays with pHrodo particles. $1 \times 10^6$ peritoneal cells were incubated with 2 μl pHrodo Red *E. coli* BioParticles for 1, 6 or 24 h to allow for phagocytosis to occur, followed by two washes with FACS buffer, surface staining with appropriate markers and fixation with PFA, as detailed above. Cells were analyzed by flow cytometry.

                                                 

**In vivo phagocytosis assay.** WT mice were injected IP with 15 μl pHrodo Red *E. coli* BioParticles together with 50 μg rCD5L or PBS. 3 h later, peritoneal cells were recovered, surface stained with appropriate markers, fixed with PFA and analyzed by flow cytometry.

**Cecal bacteria isolation, growth, and coating with rCD5L.** Bacteria were isolated from WT mice ceca, as described previously[92]. Briefly, the cecal content of 3 mice was collected, diluted in PBS and filtered through sterile gauze. The cecal suspension was diluted in brain heart infusion (BHI) medium (BD Biosciences) and incubated at 37 °C for 18 h. Afterwards, the suspension was centrifuged for 15 min at 4000 x $g$ and washed twice with PBS. The bacterial suspension was aliquoted, lyophilized, and stored at −80 °C.

A lyophilized aliquot of cecal bacteria was grown for 20 h in 100 ml BHI medium, at 37 °C with agitation. After, the bacteria were washed twice with sterile PBS to achieve an optical density (OD) at 600 nm of 2.0. For coating with rCD5L, $1 \times 10^8$ bacteria were resuspended in 500 μl TBS with 5 mM $CaCl_2$, containing 0, 2, or 10 μg rCD5L, and incubated for 1 h on ice. Then, bacteria were washed twice with PBS. The pellet was resuspended in 200 μl PBS and plated in BHI agar plates by serial dilution for CFU determination. Confirmation of binding of rCD5L to bacteria was done by western blotting. 30 μl of coated bacteria were run on SDS-PAGE and analyzed by western blotting, as detailed above. Anti-His mAb followed by HRP-conjugated antibody was used to detect rCD5L that bound bacterial cells.

**Phagocytosis assays with cecal bacteria.** Gentamycin protection assays using thioglycolate-elicited macrophages were used to enumerate intracellular bacteria. Briefly, macrophages were incubated 1 h with pre-coated cecal bacteria at a multiplicity of infection (MOI) of 5. After, cells were washed and culture media containing gentamycin was added. 1 h later, cells were washed, lysed, and CFUs enumerated by serial dilution plating.

## Cecal-ligation and puncture

Mice were subjected to surgery under isoflurane anesthesia, exposure, and ligation of the cecum (corresponding to approximately half the distance between the distal pole and the base of the cecum for medium-grade sepsis, or 75% of the distance for high-grade CLP) and through-and-through puncture with a 21 G needle. After suturing, mice were subcutaneously (sc) injected with warm 0.9% NaCl. Buprenorphine (0.08 mg/kg) was administered sc every 12 h until 48 h post-surgery. Mice were monitored twice a day.

## LPS-induced septic shock

To induce septic shock, mice were IP injected with 10 mg/kg (lethal dose) or 1.5 mg/kg (sub-lethal dose) LPS from *Escherichia coli* O111:B4 (Sigma-Aldrich).

## Therapy using recombinant mouse CD5L

2.5 or 5 mg/kg rCD5L reconstituted in PBS were administered intraperitoneally or intravenously as single or repeated doses at 3 and 6 h after CLP or LPS-induced septic shock. Control mice were injected with PBS following the same route and specific therapeutic scheme.

## Bacterial counts

For quantification of bacteria in organs, 10-fold serial dilutions of cellular suspensions were plated on BHI agar plates and grown in aerobic conditions for 18 h at 37 °C.

## Statistical analysis

All data were plotted and analyzed with GraphPad Prism software (v.10.1.1, GraphPad Software LLC). Specific statistic tests are discriminated in each figure legend and exact *P* values of relevant comparisons are shown in each graph. Only *P* values < 0.05 were considered statistically significant.

For t-SNE and LDA analysis, cytokine, CFU (blood and lung), and neutrophil data, in each treatment condition of high-grade CLP (untreated vs. IP-treated; untreated vs. IV-treated), and mid-grade CLP in CD5L⁻ vs. WT mice, were used. Data was $\log_2$ transformed and normalized by the mean value of "untreated" for each corresponding condition, and using CD5L⁻ mouse data for mid-grade CLP. t-SNE was generated in R (v. 4.1.2) using the Rtsne (v. 0.16) package with 1000 interactions, a perplexity parameter of 8, and a trade-off θ of 0.5. LDA analysis was carried out in the MASS (v. 7.3-58.2) package. t-SNE and LDA plots were generated in ggplot2 (v. 3.4). LDA vectors were scaled by 2x for better visualization.

## Reporting summary

Further information on research design is available in the Nature Portfolio Reporting Summary linked to this article.

## Data availability

RNA seq data has been deposited in European nucleotide archive (ENA) with accession number **PRJEB74080**. Other data including raw data used to generate plots and the confidence intervals when parametric tests were used to devise statistical significance are also provided in the Source Data file. Source data are provided with this paper.

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

## Acknowledgements

We thank the support of the i3S platforms Translational Cytometry and Animal Facility, and Dr. Simon Davis (University of Oxford) for insightful comments and suggestions. This work was funded by National Funds through FCT - Fundação para a Ciência e a Tecnologia, I.P., in the framework of the projects SRecognite Infect-ERA/0003/2015 and UIDB/04293/2020, to A.M.C., and LISBOA-01-0145-FEDER-030254, to M.M., and by CNRS, INSERM, the SRecognite project (ANR-Infect-ERA-2015), and the Investissement d'Avenir program PHENOMIN (French National Infrastructure for mouse Phenogenomics; ANR-10-INBS-07) to B.M. and H.L. R.F.S. and M.S.C. were recipients of PhD studentships from FCT, references SFRH/BD/110691/2015 and SFRH/BD/116791/2016, respectively. B.C. is supported by a Junior Researcher/CEEC INST21/IBMC/2802/2022.

## Author contributions

L.O., R.F.S., M.M. and A.M.C. designed and developed the *Cd5l⁻/⁻* mouse model; L.O., B.M, M.M. and A.M.C. designed the experiments; L.O., M.C.S., A.P.G., R.F.S, M.S.C., A.N. and H.L. performed the experiments; B.C. performed bioinformatics analyses; L.O., H.L., I.A., F.G., B.M, M.M. and A.M.C. analyzed biological data; L.O. and A.M.C. wrote the paper; L.O., B.M., M.M. and A.M.C. reviewed and edited the paper. All authors read and approved the manuscript.

## Competing interests

L.O. and A.M.C. are inventors on the patent application "Recombinant Human CD5L protein, active fragments or peptides derived thereof and pharmaceutical composition comprising the recombinant human CD5L protein, active fragments or peptides derived thereof for the treatment of acute infectious diseases, inflammatory diseases and sepsis", applicants, IBMC (Instituto de Biologia Molecular e Celular) and INVIGATE GmbH, WO2023/146425, published in 03.08.2023. The remaining authors declare no competing interests.
