## [Peer Review File · Nature Communications]

REVIEWER COMMENTS

Reviewer #1 (Remarks to the Author):

This is a well performed study which examines the role of CD5L in the sepsis. Using the Crispr/Cas9 system, the authors generated CD5L-deficient mice and found that CD5L-deficient mice have an impaired immune response and decreased survival to otherwise non-lethal forms of sepsis. The paper reads well, and data are convincing. The paper would be significantly improved if the authors could use single-cell transcriptomics to explore the target cells of CD5L and, in combination with other high-throughput omics techniques, elucidate the underlying mechanisms by which CD5L affects neutrophil recruitment and activation.

Major points:

1. What are the levels of CD5L in the blood or other body fluids of healthy, septic, or septic shock patients? Authors should test for CD5L levels in blood or other body fluids of patients with sepsis, or clarify this issue using publicly transcriptome data sets. Besides, the correlation between CD5L level and clinical parameters related to sepsis, such as SOFA score, PCT level, neutrophil count, should also be analyzed.
2. In Figures 1, 2, and 3, plasma ALT, AST and creatinine levels should be measured. Moreover, representative images of the liver, heart, kidney, and other organs after inducing a high-grade sepsis condition should also be presented.
3. Quantitative analysis of inflammatory genes such as TNF, IL6 and IL-1B in liver, lung and kidney tissues should be performed.
4. Does intravenous injection of recombinant CD5L (rCD5L) reverse phenotype in CD5L-deficient mice?
5. The authors suggest that CD5L plays an anti-inflammatory role without affecting the polarity of macrophages. Is it possible that CD5L affects CD4, CD8 and B cells? Single-cell transcriptomics can help clarify this question.
6. What is the mechanism by which CD5L promotes the increase of neutrophil-chemotactic CXCL1? Further exploration is recommended.
7. Although the authors propose that rCD5L promotes neutrophil recruitment and activation by increasing circulating levels of the neutrophil chemotactic agent CXCL1, this hypothesis has not been tested in vivo.

Minor issues:

1. The figure legend needs to provide and define the n number (i.e. the sample size used to derive statistics).
2. Please present a schematic diagram of this study.
3. Please provide the test results (e.g. P values) as exact values whenever possible and with confidence intervals noted.

Reviewer #2 (Remarks to the Author):

In this study, the authors investigated CD5L's therapeutic potential in sepsis using mouse models and

rCD5L administration. The results are convincing and supported by the use of both wild-type and CD5L-deficient mice, and rCD5L administration. While the authors suggest that CD5L protects against septic shock by recruiting neutrophils through CXCL1 elevation, the study's findings also align with previous evidence that CD5L can reduce inflammation caused by dead cell debris or DAMPs. Despite demonstrating CD5L's potential as a therapeutic agent against sepsis, the study requires further investigation to address concerns.

1. In lines 2-3, the authors mention the association of CD5L with IgM, but the references provided are not sufficient. To provide a more complete picture, the following articles should be included:
Arai S, et al, Obesity-associated autoantibody production requires AIM to retain the immunoglobulin M immune complex on follicular dendritic cells. *Cell Rep*, 2013, 3:1187.
Hiramoto E, et al, The IgM pentamer is an asymmetric pentagon with an open groove that binds the AIM protein. *Sci Adv*, 2018, 4: eaau1199.
2. Figures 1f, 2f, and 3f : The pictures have low resolution, making it difficult to discern the tissue status. Furthermore, if the authors like to mention quantitative differences (e.g., area of edema, wall thickness, etc.), they should perform quantification analysis to support their claims (lines 97-101 and elsewhere).
3. Lines 131-136: The author suggests that the peritoneal neutrophil enrichment was more pronounced in the IV treatment group than in the IP group. However, upon comparing the figures between 2d and 3d, it does not seem that there is a significant difference in neutrophil enrichment between the IP and IV groups. Furthermore, the increase rate between the presence and absence of rCD5L (i.e., the "enrichment") does not appear to be a critical factor in determining the prognosis. Instead, the absolute number of recruited neutrophils should be more important for supporting the survival of mice, particularly in terms of their contribution to reducing bacterial load, which the authors claim is one of the major impacts of CD5L in this model. Additionally, if the authors claim that neutrophils are more enriched in the IV treatment group, the comparison between IP and IV should be made within the experiments conducted simultaneously.
4. According to reports (including ref. 25), IgM-bound rCD5L exhibits reduced activity in capturing its target, including DAMPs. Given that IgM is abundant in the peritoneal cavity, it is likely that the injected rCD5L would immediately bind to IgM after IP, potentially leading to a loss of activity. This raises the possibility that the difference in reduction of bacteria and improved survival between IP and IV administration may be influenced by the association of rCD5L with endogenous IgM. To investigate this further, the authors should conduct western blot analysis for the peritoneal fluids in reducing conditions (without any reducing agent such as 2ME or DTT to preserve the disulfide bond between CD5L and IgM) (cf. ref. 24 or Hiramoto et al, *Sci Adv*, 2018) to determine whether the injected rCD5L is bound to IgM or not.
5. Lines 163-170: Regarding the temporal reduction of blood CD5L levels in WT mice following moderate CLP aggression, it is worth considering whether this reduction is due to the dissociation of CD5L from IgM, which allows CD5L to be excreted into urine thereby its blood level is decreased, as a result of bacterial infection. To clarify the state of CD5L, western blotting analysis (as mentioned in comment 4)

should be conducted to determine whether it is bound to IgM or not. It is unclear whether the ELISA system employed by the authors from SinoBiological detects both types of CD5L or only IgM-free CD5L. Moreover, since the main sources of CD5L in a healthy condition are peritoneal large macrophages and liver Kupffer cells, it is essential to analyze the mRNA expression changes in response to CLP stimulation. Additionally, as previously mentioned, IgM is a critical factor that determines the amount of CD5L within the body, so the level of IgM during the model should also be examined.

6. According to the lines 194-196, the observation that the IP-administration of rCD5L did not control the spreading of bacteria, which is based on the results of Fig. 2c, argues against a direct inactivation of pathogens by the recombinant protein. However, it is possible that increasing the amount of IP injected CD5L might improve its efficacy, although the authors did not investigate any dose differences in efficacy. Furthermore, the authors emphasized the increased redistribution of rCD5L to the peritoneal cavity after IV administration, but they did not investigate the difference in rCD5L levels between IP and IV in the peritoneal cavity, even though rCD5L was directly injected there in the case of IP.

7. Fig. 6h, lines 230-240: The authors mention the importance of CD5L's ability to control lethal sepsis induced by LPS, but this result is presented with only one figure and a brief mention. Given that CD5L has been reported to bind to DAMPs and suppress inflammation by phagocytosis or neutralization (ref 25), this finding is remarkable to show a physiological role of CD5L in terms of this sort of characteristic and should be emphasized more in the manuscript and added to the abstract.

Moreover, while the authors claim that neutrophil recruitment is the main function of CD5L in improving survival in the CLP-sepsis model and only mention this mechanism in the abstract, the role of CD5L in promoting the inactivation or removal of DAMPs and bacterial byproducts is also likely to contribute to the improvement of survival in the CLP-sepsis model. Therefore, the authors should investigate and discuss the role of CD5L in this aspect in sepsis further. For instance, the authors could measure changes in endotoxin levels in mice with LPS-induced sepsis or CLP-induced sepsis with or without rCD5L administration. Also, in CLP-sepsis model, the levels of types of DAMPs, in blood or tissues, are possibly measured.

Additionally, the authors only tested IP administration of rCD5L in the LPS-induced sepsis model. As the authors suggest that IV administration is more effective in controlling bacterial load in the CLP-sepsis model, it would be worthwhile to compare IP and IV administration of rCD5L in the LPS-induced sepsis model, which could provide further insights into the mechanisms of CD5L in attenuation of septic shock.

8. Regarding phagocytosis assays and bacteria killing assays. In figures 2c and 3c, the authors demonstrated that there is no difference in CFU counts in the peritoneum at 6 and 24 hours with or without rCD5L, whether administered intraperitoneally or intravenously. Although the only difference observed in CFU is in the lungs and liver with intravenous administration, the authors use peritoneal macrophages and neutrophils for the uptake and killing experiment of pHrode E. coli. The authors have been claiming from the beginning that CD5L does not play a role in events occurring within the peritoneum, so I'm wondering why the authors conducted experiments using cells from the peritoneum throughout the study. Shouldn't the same experiment be conducted using macrophages and neutrophils from the lungs or liver, where a difference was observed?

Furthermore, I am not convinced about the timing of the experiments in the series of phagocytosis assays and bacterial killing assays including figures 6a, 6b. It is possible that 30 minutes after the

introduction of bacteria, there are still many fresh macrophages or neutrophils present that could handle the incoming bacteria collectively. Is it also possible that the effect of CD5L could prevent cell exhaustion or inactivation that are caused by engulfment and digestion of bacteria and thus CD5L may impact a difference in later stages. Since CFU was examined at 6h, 24h, the authors should also look at those time points for the experiments of figures 6a, 6b .

9. Figure 7b: it is preferable to see the difference in vivo, in neutrophils isolated from mice after direct injection of pHrodo E.coli, which was performed to see macrophage in the figure 6. Why did the author conduct only in vitro experiments to see neutrophils?

10. In Figure 7c, there is only one point of information, and it is unclear how the uptake of bacteria by neutrophils changes over time (whether the bacteria decrease with time, etc.). Other time points should also be included. It may be important to examine whether there is a difference in the change rate of MFI with or without CD5L.

11. Figure 7e: How about the expression of C11b in macrophage? I'm curious about whether there is also increase of CD11b expression in macrophages (F4/80+LPM or others) or not.

12. Figure 8c: The level of CXCL1 at the healthy status should be presented as a control to show CXCL1 expression is induced along with CLP-sepsis.

13. To better understand the regulation of CXCL1 by CD5L, it would be helpful to examine whether rCD5L administration can increase CXCL1 without sepsis using healthy mice, to determine whether CXCL1 induction is specific to sepsis or can be induced in healthy conditions as well to determine whether CXCL1 induction is specific to sepsis or can be induced in healthy conditions as well.

14. To further clarify the mechanism by which CD5L induces an increase in CXCL1, additional experiments should be conducted. Firstly, the mRNA expression of CXCL1 in different tissues or cells should be examined to determine the specific types of tissues/cells impacted by administration of rCD5L. The authors discussed that the increased production of CXCL1 by CD5L is probably an indirect effect, but no experiments were performed to investigate this. Therefore, it is important to investigate the signaling pathway that leads to CXCL1 production by CD5L, which may involve different pathways in different tissues/cells.

15. The authors reported that the endotoxin level in the rCD5L used in their experiments was less than 1 EU/ μ g (equal to 1,000 EU/mg). However, this level may still be too high for in vivo administration, as the FDA recommends a maximum exposure of no more than 5 EU/kg body weight in humans. For mice, this would translate to a maximum of 62.5 EU/kg based on the human equivalent dose. Therefore, it is important for the authors to provide more information on the safety and tolerability of the rCD5L used in their study and to ensure that the dosage used does not cause any adverse effects on the animals.

16. In the CLP-model, the types of intestinal bacteria could potentially impact the outcome, making it ideal to use littermates to compare wild-type and CD5L-deficient mice. It is unclear from the manuscript whether the authors used littermates, so clarification on this matter is necessary. If littermates were not

used, the authors should explain the differences between the genotypes in bacterial profile to ensure the validity of the study's results.

Minor issues

1. It should be noted that references 1 and 13 are the same.
2. Figure 1d is missing, while 1e appears twice in the manuscript.
3. In Figure 1c, it is unclear which groups were compared using the Mann-Whitney test. Additionally, it is unclear if the comparison was made between groups at the same time point or different time points.
4. It is necessary to include information about the source, supplier, and clone name (if applicable) of the antibodies used in the study, particularly for the anti-CD5L antibody.

Reviewer #3 (Remarks to the Author):

The authors present here an elegant work evidencing a beneficial role of CD5L (an apoptosis inhibitor expressed by macrophages) in mice suffering a bacterial insult secondary to cecal ligation. This conclusion is supported by two complementary approaches using KO mice for CD5L and also a second murine model treated with exogenous recombinant CD5L, which dramatically reduced mortality compared with untreated mice. The authors perform an extensive set of assays trying to elucidate the mechanisms how CD5L mediates its protective role, assays which suggest that CD5L induces neutrophil activation and migration to the site of infection by upregulating the levels of CXCL1, helping to solve infection, down-modulating at the same time inflammation. The authors evidenced that the IV administration of rCD5L clearly have a greater biological impact and offers more benefit than the IP administration. Based on these results, the authors propose to test rCD5L in humans with sepsis.

This reviewer has a major concern: I do not think that the mice model proposed really resembles sepsis. Most human cases of sepsis occur in elderly individuals with comorbidities, which impair their ability to mount balanced responses to an infection. These individuals show also chronic endothelial dysfunction, chronic inflammation and oxidative stress. As far as I understand, the mice used in this work were healthy and young before cecal ligation. In consequence, this is a good model to reproduce an inflammatory response to a bacterial insult, which is not the same than sepsis. I would love to see how rCD5L works in elderly mice with comorbidities such as obesity, diabetes, or hypertension undergoing CLP, in example, since these models could better mimic what occurs in sepsis. It would be interesting also to reproduce the therapeutic model in immunosuppressed mice challenged with CLP, although I understand that immunosuppressed individuals represent a specific group by itself with its own peculiarities. This is one of the recommendations of MQTiPSS: an international expert consensus initiative for improvement of animal modeling in sepsis (<https://doi.org/10.1186/s40635-018-0189-y>): "Consider replication of the findings in models that include co-morbidity and/or other biological variables (i.e., age, gender, diabetes, cancer, immunosuppression, genetic background, and others)" (this reviewer is not between these authors, but I endorse this recommendation).

Many fantastic treatments working in mice have later failed in human sepsis, since they were tested basically in a pro-inflammatory animal model. This is why I think it is premature proposing this treatment to be translated directly into trials in humans.

This said, I do think that this paper suggests that CD5L could have a role in sepsis, in spite that the proposed mice model is probably not the best. I understand that repeating the assays in the conditions I suggest is probably out of the scope of the authors. I propose in consequence trying to re-approach the paper writing with the aim to reveal the role of CD5L in the acute response to a bacterial insult, which is what I think the results really support.

Minor comment:

Please avoid using the term “severe sepsis”, it is obsolete after the introduction of the new SEPSIS-3 definition in 2016

The results suggest that CD5L diminishes inflammation by inactivating and/or clearing bacterial by-products or DAMPs, but at the same time it promotes mobilization of neutrophils, which are the prototypic cells of the inflammatory response. Further clarification is needed here.

Response to Reviewers

Reviewer #1

This is a well performed study which examines the role of CD5L in the sepsis. Using the Crispr/Cas9 system, the authors generated CD5L-deficient mice and found that CD5L-deficient mice have an impaired immune response and decreased survival to otherwise non-lethal forms of sepsis. The paper reads well, and data are convincing. The paper would be significantly improved if the authors could use single-cell transcriptomics to explore the target cells of CD5L and, in combination with other high-throughput omics techniques, elucidate the underlying mechanisms by which CD5L affects neutrophil recruitment and activation.

We thank the reviewer for the careful analysis of the manuscript, and for bringing forth insightful points and valuable suggestions.

In considering the scope of our work, we respectfully note that conducting a transcriptomic analysis at the single-cell level would pose challenges, further increasing the complexity and length of an already extensive paper, potentially delaying publication significantly. On one hand, the generation of extensive data involving various cell populations from different anatomical locations makes full contextual analysis considerably more intricate; on the other, the analysis would be somewhat constrained, as mRNA findings would require confirmation at the protein level. Therefore, for consistency, we chose to explore the mechanisms of CD5L-dependent recruitment and activation of neutrophils using the methodological approach employed consistently throughout the paper.

However, we genuinely value the excellent suggestion made by the reviewer and concur that a transcriptomic analysis could provide insights into the global mechanisms underlying the role of CD5L in infection. For this purpose, we conducted a bulk RNA-seq analysis of cells recovered from the peritoneal cavity during mid-grade CLP, in WT and CD5L-deficient mice. The results of this analysis contribute to a better understanding of the role and biological functions impacted by CD5L during polymicrobial infection, and are detailed in **lines 164-200** and **Fig. 3** (new).

Major points:

1. What are the levels of CD5L in the blood or other body fluids of healthy, septic, or septic shock patients? Authors should test for CD5L levels in blood or other body fluids of patients with sepsis, or clarify this issue using publicly transcriptome data sets. Besides, the correlation between CD5L level and clinical parameters related to sepsis, such as SOFA score, PCT level, neutrophil count, should also be analyzed.

The absolute levels of CD5L in human blood have been calculated at 60 µg/ml in a recent proteomic study (Ref #2, Oskam *et al.*, 2023), and we have now indicated this information on lines 3-4. However, as CD5L is a recent marker, there are understandably many discrepancies in the literature and most publications using commercially available ELISA kits refer to basal blood levels over a wide range, between 0.1 and 5 µg/ml, so it is impossible to accurately determine the real values in the different diseases. We have chosen not to enter this discussion in the introduction of our manuscript as it is beyond the scope of the paper, and decided to refer to changes in serum CD5L levels in the different diseases as they are described in the literature, as increases or decreases.

Regarding human sepsis, Gao *et al.* report a basal serum CD5L level of < 0.1 µg/ml, that increases to ~ 0.5 µg/ml (> 5-fold) in sepsis patients (Ref #37, Gao *et al.*, 2019). We refer to this study and the correlations they have established between CD5L levels and leukocyte counts, PCT and ALT levels, and SOFA scores, in **lines 92-99**.

2. In Figures 1, 2, and 3, plasma ALT, AST and creatinine levels should be measured. Moreover, representative images of the liver, heart, kidney, and other organs after inducing a high-grade sepsis condition should also be presented.

We performed histopathological analyses of lung, liver and kidney (**Fig. S4a, b**), and measured AST and creatinine levels in WT and CD5L⁻ mice in mid-grade CLP (**Fig. S4c**), with corresponding text in **lines 154-160**.

The same histopathological analysis was performed in high-grade CLP with rCD5L therapy (**Fig. S7a, b**), and AST and creatinine levels (**Fig. S7c**), text on lines **243-247**.

We also measured ALT levels in both models using the commercially available kit from MyBioSource.com (#MBS2701479). The results obtained were, however, below the kit's detection level.

3. Quantitative analysis of inflammatory genes such as TNF, IL6 and IL-1B in liver, lung and kidney tissues should be performed.

Quantitative analysis of TNF-α, IL-6 and IL-1β genes in liver, lung and kidney was performed by RT-qPCR in WT and CD5L⁻ mice in mid-grade CLP, shown in **Fig. S4d**, text on **lines 160-163**; and in high-grade CLP with rCD5L therapy, shown in **Fig. S7d**, text on **lines 247-249**.

4. Does intravenous injection of recombinant CD5L (rCD5L) reverse phenotype in CD5L-deficient mice?

We appreciate the reviewer's suggestion regarding using rCD5L in CD5L-deficient mice to revert the phenotype. However, we hold the view that utilizing mice in experiments without therapeutic intent, or to clarify the mechanisms of therapy outside of a physiological context, might not fully align with ethical considerations. The search for therapies in CD5L-deficient mice falls into this category. For these reasons, we opted not to perform this experiment.

5. The authors suggest that CD5L plays an anti-inflammatory role without affecting the polarity of macrophages. Is it possible that CD5L affects CD4, CD8 and B cells? Single-cell transcriptomics can help clarify this question.

We conducted additional experiments to address this question, examining B1, B2 and T cell numbers after administering rCD5L via IP or IV routes (**Fig. 4c**). In exploring the *in vivo* targeting of cells by rCD5L during CLP, we now report the binding of rCD5L to a small proportion of T cells (**Fig. 9e**).

While we appreciate the suggestion for a single-cell transcriptomic analysis, its complexity poses challenges, and there is no guarantee of improved clarity. CD5L's pleotropic nature affects various pathways in different target cells, making it too intricate to analyze all involved cell subsets. It is important to note that the effects can be both direct and indirect. The complexity implies that solving this in a single step within this study is unlikely. While T cells may play a role, their involvement is not as apparent as that of neutrophils. Considering

the potential costs and the delay in publishing, pursuing further this topic to obtain uncertain information may not be justified.

6. What is the mechanism by which CD5L promotes the increase of neutrophil-chemotactic CXCL1? Further exploration is recommended.

We explored further the mechanism by analyzing which cells are targeted by CD5L, and of these which respond by increasing CXCL1 levels. Although we cannot exclude the participation of other cells by direct or indirect effects of rCD5L, we found that CD45-negative cells in the peritoneal cavity are targeted by rCD5L (**Fig. 9e**), and subsequently express increased levels of CXCL1 (**Fig. 9f**). These experiments are described in **lines 392-399**.

7. Although the authors propose that rCD5L promotes neutrophil recruitment and activation by increasing circulating levels of the neutrophil chemotactic agent CXCL1, this hypothesis has not been tested in vivo.

The experiments addressing the CD5L-dependent increase in CXCL1 production (**Fig. 9e, f, lines 392-399**) and neutrophil activation (**Fig. 9g, h, lines 401-412**) were performed *in vivo*, in the context of CLP.

Minor issues:

1. The figure legend needs to provide and define the n number (i.e. the sample size used to derive statistics).

Unless otherwise specified in the legend, all graphs now include dots representing individual subjects (= *n*) used for statistical purposes.

2. Please present a schematic diagram of this study.

We present the diagram in **Fig. 1** (new), the corresponding text is on **lines 103-106**.

3. Please provide the test results (e.g. P values) as exact values whenever possible and with confidence intervals noted.

We incorporated the exact *P*-value of the relevant comparisons in each graph. For the sake of space in the legends and clarity, we provide a supplementary excel file (**Table S5**) with the *P*-values and respective confidence intervals when parametric tests were used.

Reviewer #2

In this study, the authors investigated CD5L's therapeutic potential in sepsis using mouse models and rCD5L administration. The results are convincing and supported by the use of both wild-type and CD5L-deficient mice, and rCD5L administration. While the authors suggest that CD5L protects against septic shock by recruiting neutrophils through CXCL1 elevation, the study's findings also align with previous evidence that CD5L can reduce inflammation caused by dead cell debris or DAMPs. Despite demonstrating CD5L's potential as a therapeutic agent against sepsis, the study requires further investigation to address concerns.

We thank the reviewer for the detailed analysis of the manuscript, a series of suggestions were indeed decisive for disclosing some of the cellular mechanisms regulated by CD5L.

1. In lines 2-3, the authors mention the association of CD5L with IgM, but the references provided are not sufficient. To provide a more complete picture, the following articles should be included:

Arai S, et al, Obesity-associated autoantibody production requires AIM to retain the

immunoglobulin M immune complex on follicular dendritic cells. Cell Rep, 2013, 3:1187. Hiramoto E, et al, The IgM pentamer is an asymmetric pentagon with an open groove that binds the AIM protein. Sci Adv, 2018, 4: eaau1199.

Because the association of CD5L with IgM gained considerably more importance in the revised version of the manuscript, this topic was significantly expanded, described in the Introduction on **lines 47-56**, and including the references mentioned by the reviewer.

2. Figures 1f, 2f, and 3f : The pictures have low resolution, making it difficult to discern the tissue status. Furthermore, if the authors like to mention quantitative differences (e.g., area of edema, wall thickness, etc.), they should perform quantification analysis to support their claims (lines 97-101 and elsewhere).

We apologize to the reviewer for the quality of the images, they were submitted as a PDF file and were undoubtedly compressed in such a way that they lost quality. In any case, all histological images were replaced with new ones as we performed a more extensive and comprehensive histopathological analysis of the organs. We trust that the quality of the images in the revised version is improved to the expected standards.

To quantify differences, we performed an extended analysis using samples from 3 mice per group, measuring AST and creatinine protein levels, TNF- α , IL-6 and IL-1 β mRNA levels, and pathological scores based on the histopathological analysis of tissue sections. The latter was performed by a pathologist blinded to the experimental conditions, who classified each sample according to a four-level detailed and previously validated severity scoring system.

These analyses are described in **lines 154-163** and **Fig. S4a-d** for the mid-grade CLP model using WT and CD5L⁻ mice, and in **lines 243-249** and **Fig. S7a-d** for the high-grade CLP model with rCD5L IP or IV administration.

3. Lines 131-136: The author suggests that the peritoneal neutrophil enrichment was more pronounced in the IV treatment group than in the IP group. However, upon comparing the figures between 2d and 3d, it does not seem that there is a significant difference in neutrophil enrichment between the IP and IV groups. Furthermore, the increase rate between the presence and absence of rCD5L (i.e., the "enrichment") does not appear to be a critical factor in determining the prognosis. Instead, the absolute number of recruited neutrophils should be more important for supporting the survival of mice, particularly in terms of their contribution to reducing bacterial load, which the authors claim is one of the major impacts of CD5L in this model. Additionally, if the authors claim that neutrophils are more enriched in the IV treatment group, the comparison between IP and IV should be made within the experiments conducted simultaneously.

We recognize that the term "enrichment" may be confusing or misleading. We followed the reviewer's recommendation and clarify that the increase is in absolute number of neutrophils. This is stated in **lines 232-235** and shown in **Fig. 4c**, where the total cell number variations in IP and IV treatments now appear side by side in the same figure.

4. According to reports (including ref. 25), IgM-bound rCD5L exhibits reduced activity in capturing its target, including DAMPs. Given that IgM is abundant in the peritoneal cavity, it is likely that the injected rCD5L would immediately bind to IgM after IP, potentially leading to a loss of activity. This raises the possibility that the difference in reduction of bacteria and improved survival between IP and IV administration may be influenced by the association of rCD5L with endogenous IgM. To investigate this further, the authors should conduct western blot analysis for the peritoneal fluids in reducing conditions (without any reducing agent

such as 2ME or DTT to preserve the disulfide bond between CD5L and IgM) (cf. ref. 24 or Hiramoto et al, Sci Adv, 2018) to determine whether the injected rCD5L is bound to IgM or not.

We thank the reviewer for suggesting this line of experimentation, which led to the discovery of interesting effects we had initially overlooked. We performed western blotting analysis under non-reducing conditions to prevent disruption of intramolecular CD5L-IgM disulfide bonds, and detail the amount of IgM-complexed CD5L and of CD5L in free state in the experiments. This is shown in **Fig. 6i** and **Fig. S8e**, described in **lines 301-311** and discussed in **lines 426-438**.

5. Lines 163-170: Regarding the temporal reduction of blood CD5L levels in WT mice following moderate CLP aggression, it is worth considering whether this reduction is due to the dissociation of CD5L from IgM, which allows CD5L to be excreted into urine thereby its blood level is decreased, as a result of bacterial infection. To clarify the state of CD5L, western blotting analysis (as mentioned in comment 4) should be conducted to determine whether it is bound to IgM or not. It is unclear whether the ELISA system employed by the authors from SinoBiological detects both types of CD5L or only IgM-free CD5L. Moreover, since the main sources of CD5L in a healthy condition are peritoneal large macrophages and liver Kupffer cells, it is essential to analyze the mRNA expression changes in response to CLP stimulation. Additionally, as previously mentioned, IgM is a critical factor that determines the amount of CD5L within the body, so the level of IgM during the model should also be examined.

The ELISA system that we employ recognizes CD5L either in free or IgM-bound state (**Fig. S8c, lines 291-297**).

We extended the ELISA quantification of CD5L in blood and peritoneum to 0-72 h (**Fig. 6c, lines 275-282**) and performed the suggested western blotting analysis on these samples (**Fig. 6f, Fig. S8d, lines 295-299**).

We analyzed the mRNA expression changes in peritoneum and liver, described in **lines 265-269** and shown in **Fig. 6b**.

We also analyzed IgM levels during the model, which are shown in **Fig. 6g**, text on **lines 299-300**.

6. According to the lines 194-196, the observation that the IP-administration of rCD5L did not control the spreading of bacteria, which is based on the results of Fig. 2c, argues against a direct inactivation of pathogens by the recombinant protein. However, it is possible that increasing the amount of IP injected CD5L might improve its efficacy, although the authors did not investigate any dose differences in efficacy. Furthermore, the authors emphasized the increased redistribution of rCD5L to the peritoneal cavity after IV administration, but they did not investigate the difference in rCD5L levels between IP and IV in the peritoneal cavity, even though rCD5L was directly injected there in the case of IP.

We now quantified the amount of CD5L in the peritoneal cavity after IP or IV administration, and they were quite similar, as shown in **Fig. 6h**, text on **lines 301-305**, arguing against the better efficacy of IV treatment being a matter of dosage.

7. Fig. 6h, lines 230-240: The authors mention the importance of CD5L's ability to control lethal sepsis induced by LPS, but this result is presented with only one figure and a brief mention. Given that CD5L has been reported to bind to DAMPs and suppress inflammation by phagocytosis or neutralization (ref 25), this finding is remarkable to show a physiological role of CD5L in terms of this sort of characteristic and should be emphasized more in the manuscript and added to the abstract.

Moreover, while the authors claim that neutrophil recruitment is the main function of CD5L in improving survival in the CLP-sepsis model and only mention this mechanism in the abstract, the role of CD5L in promoting the inactivation or removal of DAMPs and bacterial byproducts is also likely to contribute to the improvement of survival in the CLP-sepsis model. Therefore, the authors should investigate and discuss the role of CD5L in this aspect in sepsis further. For instance, the authors could measure changes in endotoxin levels in mice with LPS-induced sepsis or CLP-induced sepsis with or without rCD5L administration. Also, in CLP-sepsis model, the levels of types of DAMPs, in blood or tissues, are possibly measured.

Additionally, the authors only tested IP administration of rCD5L in the LPS-induced sepsis model. As the authors suggest that IV administration is more effective in controlling bacterial load in the CLP-sepsis model, it would be worthwhile to compare IP and IV administration of rCD5L in the LPS-induced sepsis model, which could provide further insights into the mechanisms of CD5L in attenuation of septic shock.

We agree with the reviewer that this topic was underexplored, we expanded it to an entire section, “CD5L is a potent inhibitor of inflammation”, and refer to these CD5L roles in the abstract. Using both models (WT and CD5L⁻ mice in mid-grade CLP; and high-grade CLP with rCD5L therapy), we measured serum endotoxin levels and HMGB1 levels in lung, liver, and kidney, and recorded overall mouse survival. Results are shown in **Fig. 8a-g**, and described in **lines 347-374**.

We value the reviewer's suggestion for a comparative study between IP and IV administration in the LPS model. However, we believe that the fundamental principles of the therapeutic efficacy of rCD5L in this model were adequately established through IP treatment. Given the limited relevance of the LPS to direct therapeutic applications or clarifying mechanisms in medicine, and also for ethical considerations, we refrain from expanding further this line of experimentation.

8. Regarding phagocytosis assays and bacteria killing assays. In figures 2c and 3c, the authors demonstrated that there is no difference in CFU counts in the peritoneum at 6 and 24 hours with or without rCD5L, whether administered intraperitoneally or intravenously. Although the only difference observed in CFU is in the lungs and liver with intravenous administration, the authors use peritoneal macrophages and neutrophils for the uptake and killing experiment of pHrode E. coli. The authors have been claiming from the beginning that CD5L does not play a role in events occurring within the peritoneum, so I'm wondering why the authors conducted experiments using cells from the peritoneum throughout the study. Shouldn't the same experiment be conducted using macrophages and neutrophils from the lungs or liver, where a difference was observed?

Furthermore, I am not convinced about the timing of the experiments in the series of phagocytosis assays and bacterial killing assays including figures 6a, 6b. It is possible that 30 minutes after the introduction of bacteria, there are still many fresh macrophages or neutrophils present that could handle the incoming bacteria collectively. Is it also possible that the effect of CD5L could prevent cell exhaustion or inactivation that are caused by engulfment and digestion of bacteria and thus CD5L may impact a difference in later stages. Since CFU was examined at 6h, 24h, the authors should also look at those time points for the experiments of figures 6a, 6b .

It is crucial to clarify that our manuscript nowhere states or suggests that “CD5L does not play a role in events occurring within the peritoneum”. The emphasis is indeed on the peritoneum, where the primary infection and initial immune response occur. In fact, the reduction in CFUs in the peritoneum with IV-treatment highlights protective events, with a

difference of almost 3 log (2.2×10^8 to 2.9×10^5 CFU, **Fig. 4e**). We did not specifically emphasize these particular observations because the dispersion of values, influenced by common heterogeneous biological responses, results in a lack of statistical significance ($P = 0.13$). Nonetheless, the control of bacterial dissemination initiated in the peritoneal cavity will undoubtedly influence secondary responses in the other organs. We do not conduct experiments on organs such as lungs and liver because they are not the sites of initiation of infection, but mostly, injecting bacteria or bioparticles directly into these organs for the analysis of local responses by organ-resident phagocytes is completely unfeasible.

We addressed timing issues by conducting experiments collecting peritoneal cells 3 hours after introducing bioparticles, providing valuable information about the phagocytic capacity of the cells. These new findings are presented in **Fig. 7a, b** and explained in detail in **lines 318–331**. Extending the experiment beyond 3 hours would not be aligned with the main objective of the experiment, which is to evaluate the *in vivo* capacity of phagocytic cells, since the LPMs would no longer be present, or only in very reduced numbers.

9. Figure 7b: it is preferable to see the difference in vivo, in neutrophils isolated from mice after direct injection of pHrodo E.coli, which was performed to see macrophage in the figure 6. Why did the author conduct only in vitro experiments to see neutrophils?

Continuing from the previous question, we now include *in vivo* data on neutrophil phagocytosis (**Fig. 7a, b**).

10. In Figure 7c, there is only one point of information, and it is unclear how the uptake of bacteria by neutrophils changes over time (whether the bacteria decrease with time, etc.). Other time points should also be included. It may be important to examine whether there is a difference in the change rate of MFI with or without CD5L.

We thank the reviewer for this suggestion, as we see changes in neutrophil phagocytosis over time that we had overlooked. As shown in **Fig. 7e, f, lines 336-345**, although the percentage of neutrophils phagocytosing bioparticles is equivalent regardless of rCD5L, the presence of the protein results in a greater number of bioparticles being phagocytosed, increasing over time. In macrophages, however, this effect is not observed.

11. Figure 7e: How about the expression of C11b in macrophage? I'm curious about whether there is also increase of CD11b expression in macrophages (F4/80+LPM or others) or not.

We carried out this experiment and present the result here for the reviewer's consideration. There were no obvious differences in CD11b expression on macrophages between WT and CD5L⁻ mice in mid-grade CLP. Due to the considerable size of the manuscript, and because this information does not add relevant value, we chose not to include it in the revised version.

Figure 1, Rev. #2 - WT and CD5L⁻ mice were subjected to CLP surgery to induce moderate disease severity. CD11b MFI values (geo mean) in macrophages (CD45⁺CD11b⁺CD11c⁺F4/80⁺) collected from the peritoneal cavity of mice at the indicated times after surgery. Representative data from 2 independent experiments.

12. Figure 8c: The level of CXCL1 at the healthy status should be presented as a control to show CXCL1 expression is induced along with CLP-sepsis.

We now include the levels of CXCL1 at the healthy status in WT and CD5L⁻ mice, in the peritoneum and blood (**Fig. 9c**; 0 h).

13. To better understand the regulation of CXCL1 by CD5L, it would be helpful to examine whether rCD5L administration can increase CXCL1 without sepsis using healthy mice, to determine whether CXCL1 induction is specific to sepsis or can be induced in healthy conditions as well to determine whether CXCL1 induction is specific to sepsis or can be induced in healthy conditions as well.

We quantified CXCL1 levels in samples from rCD5L administration in healthy mice and present the results here for the reviewer's consideration. In the absence of disease, CXCL1 levels, if detected in some animals, are indeed minimal (NB: scale in pg/ml). As this information adds little to the article, we chose not to include it in the revised version.

Figure 11, Rev. #2 - CXCL1 concentration in the blood serum of a) naïve CD5L⁻ mice injected IV with 2.5 mg/kg rCD5L, at the indicated times after injection, and b) naïve WT mice injected IV with 2.5 or 5.0 mg/kg rCD5L or PBS (vehicle, 0 mg/kg) at 48 h after injection.

14. To further clarify the mechanism by which CD5L induces an increase in CXCL1, additional experiments should be conducted. Firstly, the mRNA expression of CXCL1 in different tissues or cells should be examined to determine the specific types of tissues/cells impacted by administration of rCD5L. The authors discussed that the increased production of CXCL1 by CD5L is probably an indirect effect, but no experiments were performed to investigate this. Therefore, it is important to investigate the signaling pathway that leads to CXCL1 production by CD5L, which may involve different pathways in different tissues/cells.

We explored further the mechanism by analyzing which cells are targeted by CD5L, and of these which respond by increasing CXCL1 levels. Although we cannot exclude the participation of other cells by indirect effects, we now show that CD5L binds directly to peritoneal CD45-negative cells (Fig. 9e), which subsequently express increased levels of CXCL1 (**Fig. 9f**). These experiments are described in **lines 392-399**.

While we appreciate the reviewer's suggestion regarding the analysis of CXCL1 mRNAs in various tissues, we find that extending the analysis to potential CD5L-target cells beyond the

peritoneal cavity introduces considerable complexity with uncertain outcomes. Although it is conceivable that organ-resident cells could elevate CXCL1, the ultimate effect might be the recruitment of neutrophils to secondary infection sites, creating ambiguity in the analyses. Given these uncertainties, and the fact that we now provide a plausible mechanism based on protein expression levels, we have opted not to pursue the suggested experiments at the mRNA level.

15. The authors reported that the endotoxin level in the rCD5L used in their experiments was less than 1 EU/μg (equal to 1,000 EU/mg). However, this level may still be too high for in vivo administration, as the FDA recommends a maximum exposure of no more than 5 EU/kg body weight in humans. For mice, this would translate to a maximum of 62.5 EU/kg based on the human equivalent dose. Therefore, it is important for the authors to provide more information on the safety and tolerability of the rCD5L used in their study and to ensure that the dosage used does not cause any adverse effects on the animals.

We obtained additional information from the producer, who stated that the endotoxin level in the samples supplied was measured at 0.21 EU/μg protein (**lines 564-566**). The safety and tolerability of rCD5L were tested in naïve WT mice intravenously injected with rCD5L. No adverse effects were observed, as shown in **Fig. S5a-c** and described in **lines 206-211**.

16. In the CLP-model, the types of intestinal bacteria could potentially impact the outcome, making it ideal to use littermates to compare wild-type and CD5L-deficient mice. It is unclear from the manuscript whether the authors used littermates, so clarification on this matter is necessary. If littermates were not used, the authors should explain the differences between the genotypes in bacterial profile to ensure the validity of the study's results.

Full information about the use of mice in the study is now provided in **lines 529-540**.

Minor issues

1. It should be noted that references 1 and 13 are the same.

Response: we thank the reviewer for spotting that error, which was now corrected.

2. Figure 1d is missing, while 1e appears twice in the manuscript.

Response: we thank the reviewer for spotting that error.

3. In Figure 1c, it is unclear which groups were compared using the Mann-Whitney test. Additionally, it is unclear if the comparison was made between groups at the same time point or different time points.

In Fig. 1c (now Fig. 2c), Mann-Whitney test was used to compare groups (WT and CD5L⁻) in each time point. We have changed the indication of *P* values to be clearer which groups were being analyzed.

4. It is necessary to include information about the source, supplier, and clone name (if applicable) of the antibodies used in the study, particularly for the anti-CD5L antibody.

We have included a new supplementary table (Table S4) containing all antibodies used in this part of the study, which were different from those used in mouse immunophenotyping.

Reviewer #3 (Remarks to the Author):

The authors present here an elegant work evidencing a beneficial role of CD5L (an apoptosis

inhibitor expressed by macrophages) in mice suffering a bacterial insult secondary to cecal ligation. This conclusion is supported by two complementary approaches using KO mice for CD5L and also a second murine model treated with exogenous recombinant CD5L, which dramatically reduced mortality compared with untreated mice. The authors perform an extensive set of assays trying to elucidate the mechanisms how CD5L mediates its protective role, assays which suggest that CD5L induces neutrophil activation and migration to the site of infection by upregulating the levels of CXCL1, helping to solve infection, down-modulating at the same time inflammation. The authors evidenced that the IV administration of rCD5L clearly have a greater biological impact and offers more benefit than the IP administration. Based on these results, the authors propose to test rCD5L in humans with sepsis.

This reviewer has a major concern: I do not think that the mice model proposed really resembles sepsis. Most human cases of sepsis occur in elderly individuals with comorbidities, which impair their ability to mount balanced responses to an infection. These individuals show also chronic endothelial dysfunction, chronic inflammation and oxidative stress. As far as I understand, the mice used in this work were healthy and young before cecal ligation. In consequence, this is a good model to reproduce an inflammatory response to a bacterial insult, which is not the same than sepsis. I would love to see how rCD5L works in elderly mice with comorbidities such as obesity, diabetes, or hypertension undergoing CLP, in example, since these models could better mimic what occurs in sepsis. It would be interesting also to reproduce the therapeutic model in immunosuppressed mice challenged with CLP, although I understand that immunosuppressed individuals represent a specific group by itself with its own peculiarities. This is one of the recommendations of MQTiPSS: an international expert consensus initiative for improvement of animal modeling in sepsis (<https://doi.org/10.1186/s40635-018-0189-y>): “Consider replication of the findings in models that include co-morbidity and/or other biological variables (i.e., age, gender, diabetes, cancer, immunosuppression, genetic background, and others” (this reviewer is not between these authors, but I endorse this recommendation).

Many fantastic treatments working in mice have later failed in human sepsis, since they were tested basically in a pro-inflammatory animal model. This is why I think it is premature proposing this treatment to be translated directly into trials in humans.

This said, I do think that this paper suggests that CD5L could have a role in sepsis, in spite that the proposed mice model is probably not the best. I understand that repeating the assays in the conditions I suggest is probably out of the scope of the authors. I propose in consequence trying to re-approach the paper writing with the aim to reveal the role of CD5L in the acute response to a bacterial insult, which is what I think the results really support. We appreciate the reviewer’s insightful reflections and valuable suggestions.

We acknowledge the limitations inherent to the mouse models used in our study and concur that CLP and LPS do not fully mirror the complexity of clinical sepsis. Indeed, like many animal models of disease, CLP and LPS serve as proxies for human sepsis and rarely replicate its full pathophysiology. CLP, for instance, represents only a fraction of sepsis cases with abdominal origin (~20%), omitting approximately half of human sepsis cases caused by lung infections. Additionally, the aggressive and lethal nature of CLP and LPS procedures induces a rapid and intense systemic pro-inflammatory response, unlike the persistent immune suppression observed in human sepsis.

Nevertheless, it is important to recognize that the CLP and LPS models cover a significant majority of scientific and preclinical studies on experimental sepsis, providing a valuable framework for comparing the efficacy of various therapeutical agents across the field. This comparative analysis was a key aspect of our study. While we did not intend to imply an immediate translation of our experimental therapy to clinical practice, we believe it holds promise for future consideration.

We endorse the recommendations of MQTiPSS, and therefore acknowledge CLP as a valid model for studying sepsis, as recognized by the panel of experts. However, in light of the reviewer's feedback, we recognize that the original manuscript could inadvertently convey the expectation for a rapid clinical translation, which is of course not feasible in the near term. Consequently, we have extensively revised the manuscript to temper the importance of our models in the context of sepsis, including an explicit statement on the limitations of the models used (**lines 107-114**). However, given that our study falls within the realm of experimental sepsis, it would be unnatural to avoid entirely referring to sepsis when discussing the CLP model.

We respectfully disagree with the reviewer's interpretation of CD5L solely as a factor controlling infection. We present several key arguments to support our position.

1 - Transcriptomic profiling, conducted as part of our study, reveals an exaggerated inflammatory response consistent with the complex and dysregulated immune response observed in sepsis. This suggests that CD5L may play a broader role beyond simply controlling infection.

2 - Our data demonstrate that IV treatment significantly reduces CFUs in both the peritoneum and blood by more than a 2-log difference (**Fig. 4e**). This effect is not observed with IP treatment, despite equivalent CD5L levels in both cases. This implies that CD5L's function is not limited to pathogen inactivation or control of bacterial spreading.

3 - The study by Gao *et al.* (Ref #63, 2019) injects rCD5L concurrently with CLP induction, resulting in increased mortality. This outcome contradicts the notion that CD5L solely possesses antimicrobial properties. Rather, it suggests that by suppressing inflammation, rCD5L administered during the early stages of disease initiation may dampen the natural beneficial immune response, thus adversely affecting clinical outcomes. In our protocol, however, rCD5L administration occurs later, after the initial inflammatory phase, aiming to mitigate the uncontrolled systemic inflammation without interfering with the early immune response.

These points collectively suggest that CD5L's role extends beyond mere infection control and encompasses modulation of the inflammatory response, highlighting its potential as a multifaceted therapeutic agent in sepsis.

Minor comment:

Please avoid using the term "severe sepsis", it is obsolete after the introduction of the new SEPSIS-3 definition in 2016

We thank the reviewer for the constructive criticism, we abstained from using this term.

The results suggest that CD5L diminishes inflammation by inactivating and/or clearing bacterial by-products or DAMPs, but at the same time it promotes mobilization of neutrophils, which are the prototypic cells of the inflammatory response. Further clarification is needed here.

The role of neutrophils in limiting inflammation in the context of sepsis, and in our models after the administration of rCD5L, is discussed in **lines 484-498**.

REVIEWERS' COMMENTS

Reviewer #1 (Remarks to the Author):

Remarks to the Author:

1. For the minor issues 2 mentioned in my previous comment, where I requested the inclusion of a schematic diagram in the study, it seems there might have been some confusion. Specifically, I recommended that the author provide a schematic diagram illustrating the proposed mechanism hypothesis, rather than a study design flow chart. You may refer to the graphical abstracts in the following references for guidance.

[1] Nascimento DC, Viacava PR, Ferreira RG, Damaceno MA, Piñeros AR, Melo PH, Donate PB, Toller-Kawahisa JE, Zoppi D, Veras FP, Peres RS, Menezes-Silva L, Caetité D, Oliveira AER, Castro ÍMS, Kauffenstein G, Nakaya HI, Borges MC, Zamboni DS, Fonseca DM, Paschoal JAR, Cunha TM, Quesniaux V, Linden J, Cunha FQ, Ryffel B, Alves-Filho JC. Sepsis expands a CD39⁺ plasmablast population that promotes immunosuppression via adenosine-mediated inhibition of macrophage antimicrobial activity. *Immunity*. 2021 Sep 14;54(9):2024-2041.e8. doi: 10.1016/j.immuni.2021.08.005. Epub 2021 Sep 1. PMID: 34473957.

[2] Chen X, Wu R, Li L, Zeng Y, Chen J, Wei M, Feng Y, Chen G, Wang Y, Lin L, Luo H, Chen A, Zeng Z, He F, Bai Y, Zhang S, Han Y, Wang Z, Zhao X, Xiao W, Jiang Y, Gong S. Pregnancy-induced changes to the gut microbiota drive macrophage pyroptosis and exacerbate septic inflammation. *Immunity*. 2023 Feb 14;56(2):336-352.e9. doi: 10.1016/j.immuni.2023.01.015. PMID: 36792573.

2. Please align the graphical representation of data in Figures S3 and S6 with the style used in Figures 2E and 4D. This adjustment would enhance the overall uniformity of the manuscript.

3. The authors mentioned on line 249 that “with the only exception of IL-1b which was increased in the kidney 24 h after IV treatment (Fig. S7d).” However, the renal IL-1B levels in Figure S7d do not appear to be marked with statistical significance. Is this due to a labeling oversight, or is there indeed no statistically significant difference?

Reviewer #2 (Remarks to the Author):

The Authors have adequately addressed my concerns, and I appreciate their efforts to provide additional information that supports their conclusions. However, there are some minor remaining issues that I would like to highlight:

1. In lines 43-44, the authors have newly added the information that the blood concentration of CD5L is 60 µg/mL. However, the values appear to differ in various reports. For example, according to references #14 (Arai, *Cell Rep*, 2014); Yamazaki T, *PLoS One* 2014, 9:e109123; Koyama N, *J Gastroenterol* 2018, 53:770-779; Yasuda K, *J Autoimmun* 2023, 142:103149, the serum levels of CD5L in both humans and mice typically range from 1-20 µg/mL, with an approximate average of 5 µg/mL in humans. The ELISA systems that were independently developed are currently distributed, and some of them are not properly validated. Additionally, they often do not clearly mention whether only the free form or IgM-bound form of CD5L, or total CD5L, could be measured. Thus, defining the level of CD5L as 60 µg/mL in

this article could be considered risky. Indeed, the authors show in their figures S1 and elsewhere that the serum level of CD5L in wild-type mice as around 1000 ng/mL, which is far from 60 µg/mL. A more careful description would be preferable.

2. Regarding the material for mouse recombinant CD5L in the method section (line 564), it is preferable to describe at least the information of host cells and, if possible, the method of purification, even though it was obtained commercially. This is because whether the host cells are mammalian cells, bacterial cells, or insect cells could greatly affect the activity of CD5L.

Response to Reviewers

Reviewer #1:

1. For the minor issues 2 mentioned in my previous comment, where I requested the inclusion of a schematic diagram in the study, it seems there might have been some confusion. Specifically, I recommended that the author provide a schematic diagram illustrating the proposed mechanism hypothesis, rather than a study design flow chart. You may refer to the graphical abstracts in the following references for guidance.

*[1] Nascimento DC, Viacava PR, Ferreira RG, Damaceno MA, Piñeros AR, Melo PH, Donate PB, Toller-Kawahisa JE, Zoppi D, Veras FP, Peres RS, Menezes-Silva L, Caetité D, Oliveira AER, Castro ÍMS, Kauffenstein G, Nakaya HI, Borges MC, Zamboni DS, Fonseca DM, Paschoal JAR, Cunha TM, Quesniaux V, Linden J, Cunha FQ, Ryffel B, Alves-Filho JC. Sepsis expands a CD39⁺ plasmablast population that promotes immunosuppression via adenosine-mediated inhibition of macrophage antimicrobial activity. *Immunity*. 2021 Sep 14;54(9):2024-2041.e8. doi: 10.1016/j.immuni.2021.08.005. Epub 2021 Sep 1. PMID: 34473957.*

*[2] Chen X, Wu R, Li L, Zeng Y, Chen J, Wei M, Feng Y, Chen G, Wang Y, Lin L, Luo H, Chen A, Zeng Z, He F, Bai Y, Zhang S, Han Y, Wang Z, Zhao X, Xiao W, Jiang Y, Gong S. Pregnancy-induced changes to the gut microbiota drive macrophage pyroptosis and exacerbate septic inflammation. *Immunity*. 2023 Feb 14;56(2):336-352.e9. doi: 10.1016/j.immuni.2023.01.015. PMID: 36792573.*

We apologize for any misunderstanding. We have now created a schematic diagram of the proposed mechanism, which is now represented as Figure 10.

2. Please align the graphical representation of data in Figures S3 and S6 with the style used in Figures 2E and 4D. This adjustment would enhance the overall uniformity of the manuscript.

We thank the reviewer for their comment. However, it is important to note that the datasets represented in the Main Figures (now Figs. 1e and 3d) differ in nature from those illustrated in the Supplementary Figures (now Figs. S4 and S7), as the former are ratios derived from the latter. Consequently, presenting them in the same graphical style is impractical. Even if we aimed for uniformity in graphical representation for aesthetic purposes, this wouldn't be advisable. This is because Figures 1e and 3d utilize color gradients to represent averaged values, whereas Figures S4 and S7 depict the natural distribution of values, providing a more realistic perspective on the natural variation within the biological data.

3. The authors mentioned on line 249 that “with the only exception of IL-1b which was increased in the kidney 24 h after IV treatment (Fig. S7d).” However, the renal IL-1B levels in Figure S7d do not appear to be marked with statistical significance. Is this due to a labeling oversight, or is there indeed no statistically significant difference?

We thank the reviewer for flagging that error. The renal IL1 β mRNA expression is indeed elevated in the IV-treated group at 24 hours. However, due to the value being borderline ($P = 0.0497$), the software rounded it up to 0.05. Consequently, it was interpreted as no longer being less than 0.05, and thus deemed not significant. We have since updated the software to recognize three significant digits, enhancing its accuracy in determining significance. The text has been corrected accordingly (lines 251-254 and updated Fig. S9).

Reviewer #2:

The Authors have adequately addressed my concerns, and I appreciate their efforts to provide additional information that supports their conclusions. However, there are some minor remaining issues that I would like to highlight:

1. In lines 43-44, the authors have newly added the information that the blood concentration of CD5L is 60 µg/mL. However, the values appear to differ in various reports. For example, according to references #14 (Arai, Cell Rep, 2014); Yamazaki T, PLoS One 2014, 9:e109123; Koyama N, J Gastroenterol 2018, 53:770-779; Yasuda K, J Autoimmun 2023, 142:103149, the serum levels of CD5L in both humans and mice typically range from 1-20 µg/mL, with an approximate average of 5 µg/mL in humans. The ELISA systems that were independently developed are currently distributed, and some of them are not properly validated. Additionally, they often do not clearly mention whether only the free form or IgM-bound form of CD5L, or total CD5L, could be measured. Thus, defining the level of CD5L as 60 µg/mL in this article could be considered risky. Indeed, the authors show in their figures S1 and elsewhere that the serum level of CD5L in wild-type mice as around 1000 ng/mL, which is far from 60 µg/mL. A more careful description would be preferable.

We agree with the reviewer's concern about the challenge of defining a precise level of CD5L concentration in serum due to the variability among ELISA kits. However, this variability impedes cross-comparison between studies conducted under different conditions, underscoring the importance of establishing a reference value using the most reliable experimental method available. We have amended the text to caution that values obtained in a specific study may not be directly comparable with those from other studies (lines 45-50).

2. Regarding the material for mouse recombinant CD5L in the method section (line 564), it is preferable to describe at least the information of host cells and, if possible, the method of purification, even though it was obtained commercially. This is because whether the host cells are mammalian cells, bacterial cells, or insect cells could greatly affect the activity of CD5L.

We have provided the relevant information regarding the protein source, production and purification, in lines 571-577.